# Palmitic acid in type 2 diabetes mellitus promotes atherosclerotic plaque vulnerability via macrophage Dll4 signaling

Xiqiang Wang[1,5], Ling Zhu[1,5], Jing Liu[1], Yanpeng Ma[1], Chuan Qiu [2], Chengfeng Liu[1], Yangchao Gong[1], Ya Yuwen[1,3], Gongchang Guan[1], Yong Zhang[1] ✉, Shuo Pan [1] ✉, Junkui Wang [1] ✉ & Zhongwei Liu [1,4] ✉

Patients with Type 2 Diabetes Mellitus are increasingly susceptible to atherosclerotic plaque vulnerability, leading to severe cardiovascular events. In this study, we demonstrate that elevated serum levels of palmitic acid, a type of saturated fatty acid, are significantly linked to this enhanced vulnerability in patients with Type 2 Diabetes Mellitus. Through a combination of human cohort studies and animal models, our research identifies a key mechanistic pathway: palmitic acid induces macrophage Delta-like ligand 4 signaling, which in turn triggers senescence in vascular smooth muscle cells. This process is critical for plaque instability due to reduced collagen synthesis and deposition. Importantly, our findings reveal that macrophage-specific knockout of Delta-like ligand 4 in atherosclerotic mice leads to reduced plaque burden and improved stability, highlighting the potential of targeting this pathway. These insights offer a promising direction for developing therapeutic strategies to mitigate cardiovascular risks in patients with Type 2 Diabetes Mellitus.

Acute vascular events are considerably more prevalent in patients with type 2 diabetes mellitus (T2DM) than those without the disease, which contributes to the increased risk of cardiovascular complications in this population[1]. A strong association has been observed between acute vascular events and atherosclerotic plaque vulnerability which is also correlated with T2DM[2,3], emphasizing the importance of understanding the underlying mechanisms of plaque instability in the context of T2DM. The exact causes of increased plaque vulnerability in T2DM patients remain unclear, despite progress in understanding T2DM-related atherosclerosis. Uncovering the contributing factors and mechanisms in T2DM patients is vital for targeted therapies to reduce acute vascular events and enhance patient outcomes.

Patients with type 2 diabetes often exhibit significant metabolic disorders, which contribute to the increased risk of cardiovascular complications[4]. Among these disturbances, alterations in fatty acid metabolism play a pivotal role, particularly the accumulation of saturated fatty acids (SFAs)[5]. Palmitic acid (PA, C16:0), the most abundant saturated fatty acid in circulation, has been closely associated with both plaque vulnerability and major adverse cardiovascular events in T2DM[6,7]. Thus, the association between palmitic acid and increased cardiovascular risk in T2DM patients is thought to be mediated, at least in part, by its impact on atherosclerotic plaque instability.

Macrophage M1-type polarization has been identified as a significant factor contributing to plaque instability in atherosclerosis[8]. According to our and others' previous investigations, M1 polarized

[1]Department of Cardiology, Shaanxi Provincial People's Hospital, Xi'an, Shaanxi Province 710068, China. [2]Division of Bioinformatics and Genomics, Deming Department of Medicine, Tulan Center of Biomedical Informatics and Genomics, Tulane University, New Orleans, LA 70112, USA. [3]Medical School, Xizang Minzu University, Xianyang, Shaanxi Province 712000, China. [4]Affiliated Shaanxi Provincial People's Hospital, Medical Research Institute, Northwestern Polytechnical University, Xi'an, Shaanxi Province 710072, China. [5]These authors contributed equally: Xiqiang Wang, Ling Zhu. ✉e-mail: zhangyong971292@163.com; panshuosx@163.com; junkuiwang@yeah.net; medicalman@163.com

macrophages exhibit a pro-inflammatory phenotype and are characterized by the high expression of Delta-like ligand 4 (Dll4), a ligand for the Notch signaling pathway[9–11]. Consequent activation of the Notch signaling pathway in vascular smooth muscle cells (VSMCs) has been shown to induce cellular senescence, which can negatively impact the stability of atherosclerotic plaques[11,12]. Senescent VSMCs exhibit a reduced capacity for repair and extracellular matrix remodeling, leading to increased vulnerability of the plaque to rupture and subsequent acute vascular events[13]. Therefore, exploring the connection between M1 macrophage polarization, Notch signaling, and VSMC senescence could reveal key mechanisms of plaque instability in T2DM.

In this study, we embarked on a comprehensive investigation to unravel the complex interplay between metabolic disorders in T2DM and atherosclerotic plaque vulnerability. First, we examined the role of palmitic acid accumulation, a characteristic feature of metabolic disorders in T2DM, and its potential impact on plaque vulnerability. Next, we delved into the relationship between

elevated palmitic acid levels and the plaque instability phenotype, focusing particularly on the role of hyper-expression of Dll4 during M1 macrophage polarization. Thirdly, we specifically examined how Dll4-expressing macrophages could potentially induce VSMC senescence through the Notch pathway, contributing to plaque vulnerability.

## Results

### T2DM increased the risk of acute cerebral-cardiovascular events

In this study, we scrutinized the correlation between Type 2 Diabetes Mellitus (T2DM) and acute cerebro-cardiovascular incidents, primarily cerebral infarction and acute coronary syndrome, which frequently originate from the rupture of vulnerable atherosclerotic plaques[14]. We drew upon data from the National Inpatient Sample (NIS) database, encompassing 18,173,885 patients, of which 4,592,091 were diagnosed with T2DM, and 13,581,794 were without T2DM (Fig. 1A).

The T2DM cohort skewed towards an older demographic (mean age: 66.69 vs. 54.90; P < 0.001) and a lower female

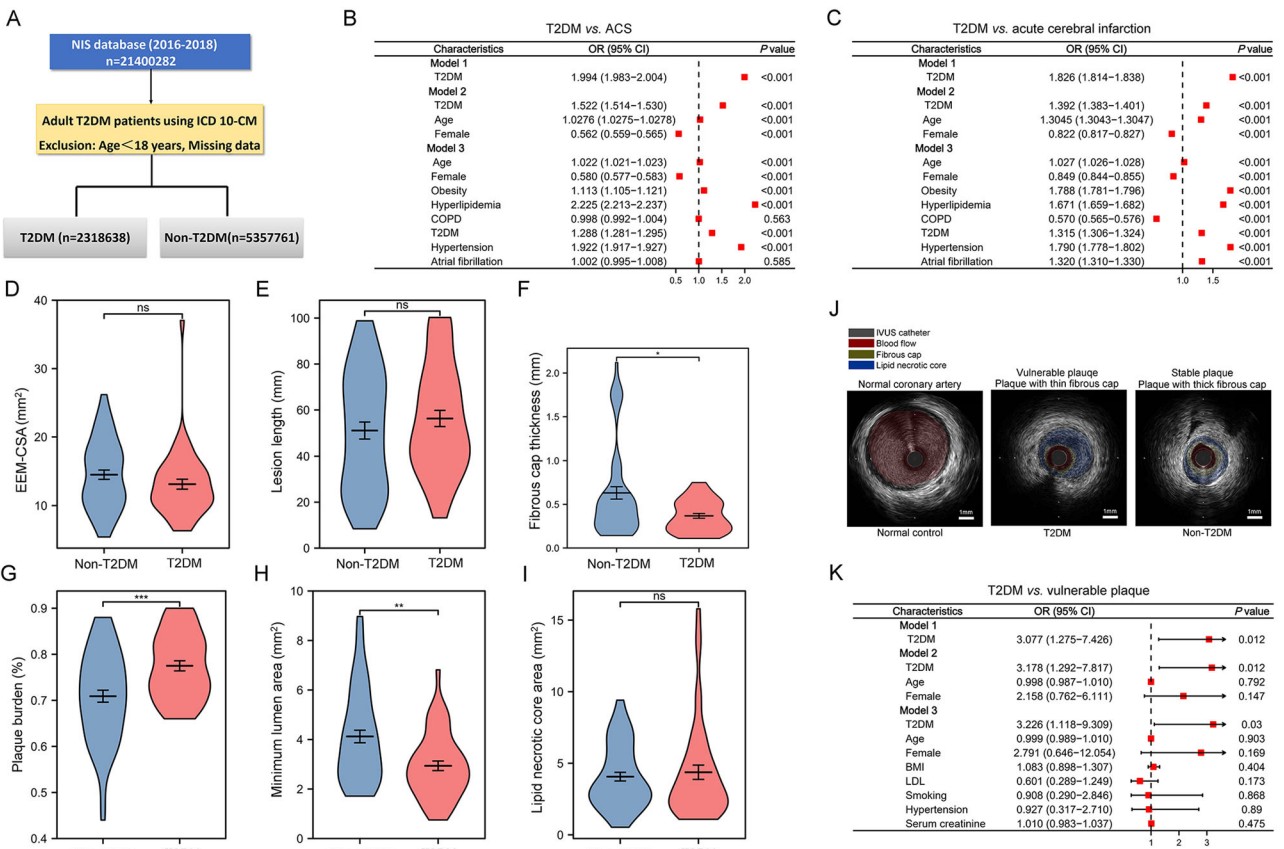

**Fig. 1 | Interplay between type 2 diabetes mellitus (T2DM), acute vascular events, and plaque vulnerability. A** Schematic representation of the cohort study derived from the National Inpatient Sample (NIS) database. **B, C** Logistic regression analysis depicting the relationship between T2DM and ACS (**B**), and T2DM and acute cerebral infarction (**C**) in the NIS database (2016-2018). Model 1 indicates the univariate regression analysis; Model 2 is adjusted for age, gender, and T2DM; Model 3 is further adjusted for additional confounders including obesity, hyperlipidemia, COPD (chronic obstructive pulmonary disease), hypertension, and atrial fibrillation. (Error bars, which extend from the center of the red square, represent the 95% confidence interval for odds ratio. The center for the error bars is the point estimate of the odds ratio for each variable. n = 21400282 subjects).

**D–I** Comparative analyses of various plaque and vascular characteristics, including external elastic membrane (EEM) - cross-sectional area (CSA). (**D**, data presented as mean ± SD, P = 0.1678), lesion length (**E**, data presented as mean ± SD, P = 0.3074), fibrous cap thickness (**F**, data presented as mean ± SD, *P = 0.017), plaque burden

(**G**, data presented as mean ± SD, ***P = 0.0002), minimum lumen area (**H**, data presented as mean ± SD, **P = 0.005), and lipid necrotic core area (**I**, data presented as mean ± SD, P = 0.6005), between non-T2DM and T2DM subjects.

**J** Representative IVUS (intravascular ultrasound) images of a normal coronary artery, a vulnerable plaque, and a stable plaque. The IVUS catheter, blood flow, fibrous cap, and lipid necrotic core are indicated in brown, red, yellow, and blue, respectively. **K** Logistic regression analysis revealing the relationship between T2DM and plaque vulnerability in the cohort study. Model 1 indicates the univariate regression analysis; Model 2 is adjusted for age, gender, and T2DM; Model 3 is further adjusted for T2DM, age, gender, BMI, LDL, smoking, hypertension, and serum creatinine levels. (Error bars, which extend from the center of the red square, represent the 95% confidence interval for odds ratio. The center for the error bars is the point estimate of the odds ratio for each variable. n = 21400282 subjects.

**D, E** Two-tailed unpaired Student t-test, **F, G, H, I** Two-tailed unpaired Student t-test with Welch's correction). Source data are provided as a Source Data file.

## Table 1 | Demographics of patients with and without T2DM in National Inpatient Sample (NIS) 2016-2018

| Characteristic | non-T2DM (n = 13581794) | T2DM (n = 4592091) | P Value* |
|---|---|---|---|
| Age, yrs (mean ± SD) | 54.90 ± 21.24 | 66.69 ± 13.75 | <0.001 |
| Female, % | 8220410 (60.5) | 2272628 (49.5) | <0.001 |
| **Race** | | | <0.001 |
| White, % | 8993332 (68.7) | 2824445 (63.3) | |
| African American, % | 1857060 (14.2) | 802082 (18.0) | |
| Hispanic, % | 1415053 (10.8) | 541299 (12.1) | |
| Asian/Pacific Islander, % | 355142 (2.7) | 126612 (2.8) | |
| Native American, % | 74620 (0.6) | 35807 (0.8) | |
| Other races, % | 397865 (3.0) | 129260 (2.9) | |
| **Comorbidity** | | | |
| Coagulopathy, % | 246117 (1.8) | 90643 (2.0) | <0.001 |
| Obesity, % | 1587910 (11.7) | 1203859 (26.2) | <0.001 |
| Hypertension, % | 4260777 (31.4) | 2019451 (44.4) | <0.001 |
| Hypothyroidism, % | 1466030 (10.8) | 712783 (15.5) | <0.001 |
| Coronary artery disease, % | 2027255 (14.9) | 1665612 (36.3) | <0.001 |
| Atrial fibrillation, % | 1695226 (12.5) | 967253 (21.1) | <0.001 |
| Cerebral infarction, % | 244754 (1.8) | 148894 (3.2) | <0.001 |
| Cerebral hemorrhage, % | 78968 (0.6) | 32323 (0.7) | <0.001 |
| Peripheral vascular disease, % | 169118 (1.2) | 119252 (2.6) | <0.001 |
| Hypercholesteremia, % | 2940357 (21.6) | 2288053 (49.8) | <0.001 |
| Alcohol use, % | 915826 (6.7) | 162651 (3.5) | <0.001 |
| Tobacco abuse, % | 2546580 (18.7) | 1146929 (25.0) | <0.001 |
| In-hospital mortality, % | 281187 (2.1) | 122824 (2.7) | <0.001 |
| Length of hospital stay, days | 4.52 ± 6.26 | 5.39 ± 6.55 | <0.001 |
| Total charges, US$ | 50819.45 ± 88285.42 | 61719.44 ± 89851.24 | <0.001 |
| ACS, % | 374223 (2.8) | 245535 (5.3) | <0.001 |
| UA, % | 22434 (0.16) | 11297 (0.25) | <0.001 |
| STEMI, % | 87929 (0.65) | 41252 (0.9) | <0.001 |
| NSTEMI, % | 264830 (1.9) | 193033 (4.2) | <0.001 |
| Cerebral infarction, % | 244754 (1.8) | 148894 (3.2) | <0.001 |

*ACS* acute coronary syndrome, *UA* unstable angina, *STEMI* ST elevated myocardial infarction, *NSTEMI* non- ST elevated myocardial infarction.

*P values were calculated using Chi-Squared tests for categorical variables, and Student's t-tests for all other continuous variables. All statistical tests are two-sided.

## Table 2 | Baseline characteristics of the patients with stable coronary artery heart disease

| Characteristics | Non-T2DM (n = 55) | T2DM (n = 46) | P value* |
|---|---|---|---|
| Age (years) | 61.49 ± 10.78 | 60.41 ± 13.378 | 0.332 |
| Male (%) | 43 (78.2) | 37 (80.4) | 0.781 |
| BMI (kg/m$^2$) | 24.78 ± 3.40 | 24.30 ± 2.80 | 0.446 |
| Hypertension (%) | 34 (61.8) | 29 (63) | 0.899 |
| Current smoker (%) | 36 (65.5) | 29 (63) | 0.801 |
| Former smoker (%) | 7 (12.7) | 6 (13) | 0.763 |
| Never smoker (%) | 12 (21.8) | 11 (23.9) | 0.653 |
| **Baseline medications (%)** | | | |
| Statin use (%) | 54 (98.2) | 46 (100) | 0.358 |
| Beta-blocker use (%) | 42 (76.4) | 34 (73.9) | 0.592 |
| Aspirin use (%) | 53 (96.4) | 43 (93.5) | 0.459 |
| ACE inhibitor use (%) | 45 (81.8) | 38 (82.6) | 0.918 |
| P2Y12 inhibitor (%) | 54 (98.2) | 46 (100) | 0.358 |
| **Baseline parameters** | | | |
| NT-proBNP (ng/L) | 667.34 ± 1502.72 | 611.07 ± 1010.87 | 0.843 |
| LDL-cholesterol (mmol/L) | 2.39 ± 0.77 | 2.58 ± 2.91 | 0.645 |
| HDL-cholesterol (mmol/L) | 1.02 ± 0.20 | 0.99 ± 0.26 | 0.622 |
| Triglyceride (mmol/L) | 1.45 ± 0.66 | 1.48 ± 1.02 | 0.822 |
| Systolic blood pressure (mmHg) | 132.00 ± 21.83 | 135.52 ± 17.66 | 0.381 |
| Diastolic blood pressure (mmHg) | 77.85 ± 11.45 | 94.41 ± 99.81 | 0.225 |
| **Echocardiographic data** | | | |
| LV EDD (mm) | 34.57 ± 7.41 | 34.69 ± 7.64 | 0.940 |
| LV ESD (mm) | 48.89 ± 6.18 | 48.91 ± 6.59 | 0.986 |
| LVEF (%) | 57.48 ± 7.56 | 55.69 ± 11.55 | 0.356 |

*ACE* angiotensin-converting enzyme, *BMI* body mass index, *CRP* C-reactive protein, *HDL* high-density lipoprotein, *LDL* low-density lipoprotein, *NT-pro BNP* N-terminal B-type natriuretic peptide, *LVEDD* left ventricular end-diastolic diameter, *LVESD* left ventricular end-systolic diameter, *LVEF* left ventricular ejection fraction. Values are percent or means ± SD.

*P values were calculated using Chi-Squared tests for categorical variables, and Student's t-tests for all other continuous variables. All statistical tests are two-sided.

### T2DM was strongly associated with characters of vulnerable atherosclerosis plaques

In our study, we conducted an in-depth examination of plaque characteristics using intravascular ultrasound (IVUS) in patients who underwent invasive coronary angiography for the diagnosis of chronic stable angina pectoris in our Chronic Stable Coronary Atherosclerotic Disease Prospective Cohort as detailed in Fig. S1. Baseline clinical parameters of participants, encapsulating demographic data, medication history, blood biochemistry, and echocardiographic data, are collated in Table 2. There was no significant difference between T2DM and non-T2DM patients in these parameters.

IVUS volumetric analysis across the entire imaged segments revealed several notable differences between the T2DM and non-T2DM cohorts (Fig. 1D–I). T2DM patients were characterized by thinner fibrous caps (*P* < 0.001, Fig. 1F), increased plaque burden (*P* < 0.001, Fig. 1G), and a smaller minimum lumen area (*P* < 0.01, Fig. 1H). However, parameters such as external elastic membrane cross-sectional area (EEM-CSA), lesion length, and the lipid necrotic core area did not vary significantly between the groups (Fig. 1D, E, and I). Representative IVUS images depicting normal coronary arteries, vulnerable plaques, and stable plaques are illustrated in Fig. 1J.

Further analysis using univariate logistic regression substantiated that T2DM was a significant determinant of "greater plaque burden", a

representation (49.5% *vs.* 60.5%; *P* < 0.001). Moreover, T2DM patients exhibited a higher incidence of cerebral infarction (3.20% *vs.* 1.80%), unstable angina (UA) (0.25% *vs.* 0.16%; *P* < 0.001), ST-segment elevation myocardial infarction (STEMI) (0.9% *vs.* 0.65%; *P* < 0.001), and non-ST segment elevation myocardial infarction (NSTEMI) (4.2% *vs.* 1.9%; *P* < 0.001). These findings were corroborated by a notable rise in in-hospital mortality (2.7% *vs.* 2.1%; *P* < 0.001), extended hospital stay duration (5.39 *vs.* 4.52 days; *P* < 0.001), and elevated total medical costs (USD 61,719.44 *vs.* 50,819.45; *P* < 0.001) among T2DM patients (Table 1).

Univariate logistic regression analysis further substantiated T2DM as a significant determinant for acute coronary syndrome (ACS) [OR 1.994 (1.983–2.004); *P* < 0.001] and acute cerebral infarction [OR 1.826 (1.814–1.838); *P* < 0.001] (Fig. 1B, C). After controlling for potential confounding risk factors - age, gender, obesity, hyperlipidemia, chronic obstructive pulmonary disease (COPD), hypertension, and atrial fibrillation, T2DM maintained a strong association with ACS [OR 1.288 (1.281–1.295); *P* < 0.001] and acute cerebral infarction [OR 1.315 (1.306–1.324); *P* < 0.001].

characteristic of "vulnerable plaque" [OR 3.07 (1.275-7.426); $P = 0.012$]. After adjustment for other confounding risk factors such as age, gender, BMI (body mass index), LDL (low-density lipoprotein), smoking, hypertension, and serum creatinine, T2DM continued to associate strongly with "vulnerable plaque" [OR 3.226 (1.118-9.309); $P = 0.003$] (Fig. 1K).

## Overall metabolomics profiling and identification of metabolic alterations in collected human samples in cohort

To guarantee the reliability and stability of our metabolomic analysis, rigorous data quality control (QC) was conducted via Pearson correlation analysis. The high Pearson correlation established across negative and positive electrospray ionization modes (ESI- and ESI+, respectively) and the combined mode (COMB, where duplicated data was removed) endorses the quality and consistency of our data (Fig. S3A-C). In the process, we identified 1969, 1982, and 3951 annotated metabolites in ESI-, ESI+ and COMB respectively.

Employing orthogonal partial least squares-discriminant analysis (OPLS-DA) score plots, we discerned distinct metabolite clusters in individuals with T2DM compared to those without, in the ESI- (Fig. S4A), and ESI+ (Fig. S4B), and COMB (Fig. 2A). Subsequently, we constructed an OPLS-DA model incorporating metabolomic data, and its ability to accurately categorize new samples was assessed through seven-fold cross-validation with 200 random permutation tests. This exercise substantiated the model's reliability and avoided overfitting for the ESI- (Fig. S4C), ESI+ (Fig. S4D), and COMB (Fig. 2B).

Volcano plots were used to depict fold changes in the levels of annotated metabolites in T2DM relative to non-T2DM, taking into account the statistically significant difference in variable importance (VIP) in the projection. The levels of these differential metabolites in T2DM significantly diverged from those in non-T2DM in the ESI- (Fig. S4E), ESI+ (Fig. S4F), and COMB (Fig. 2C).

Our analysis aimed to identify metabolic phenotypes potentially associated with coronary atherosclerosis in the context of T2DM. To this end, we compared metabolic changes observed between non-T2DM and T2DM individuals. This comparison identified variations in 410, 522, and 955 significantly changed (T2MD vs. non-T2DM, $P < 0.05$) metabolites within the ESI-, ESI+, and COMB respectively. An extensive overlap of 99.92% was observed in the annotated metabolites between non-T2DM and T2DM groups (Fig. 2D), suggesting shared metabolic shifts potentially implicated in T2DM-associated coronary atherosclerosis. As demonstrated in Fig. 2E, the Log2 fold change of the top 30 significantly changed metabolites in COMB were plotted.

## Delineating Distinct Metabolomic Patterns in T2DM Versus Non-T2DM Subjects within the Cohort

To gain insights into the functional features of the identified altered metabolites, comprehensive databases including the Kyoto Encyclopedia of Genes and Genomes compound database (KEGG), Human Metabolome Database (HMDB), and Lipid Maps were employed for detailed annotation and pathway analysis. Analysis using KEGG annotation revealed that pathways related to "lipid metabolism" and "amino acid metabolism" were significantly perturbed in the serum metabolites of T2DM subjects within the chronic stable angina cohort (Fig. S5A-C). Further, HMDB annotation identified "Lipids and lipid-like molecules" and "Organoheterocyclic compounds" as the most significantly altered functional classes (Fig. S5D-F). Given that these pathways and classes are intrinsically linked to lipid metabolism disturbances, further exploration focused on lipid metabolic alterations via the Lipid Maps database. As presented in Fig. 2F and Table S4, the frequency of annotated metabolites in the "Fatty acids and conjugates" class under the category of "Fatty acyls" exhibited a marked frequency of annotations. In our analysis, the category "fatty acyls" and class "fatty acid and conjugates" showed a notably higher frequency of significant metabolic alterations compared to other categories and

classes, as illustrated in Fig. 2G-H. Fig. 2I presents the Log2 fold changes of the significantly altered metabolites within the "fatty acid and conjugates" class. Notably, specific saturated fatty acids, such as icosanoic acid, tetracosanedioic acid, palmitic acid, and ximenic acid, exhibited differential levels in T2DM subjects relative to their non-T2DM counterparts in the cohort.

## Metabolomic profiling and metabolic alteration identification in animal models

To parallel the pathophysiology observed in the human cohort, we established mouse models of atherosclerosis (AS) and atherosclerosis complicated by T2DM (AS + DM) (Fig. 3A). Successful induction of diabetes was confirmed by a significant increase in fasting blood glucose concentration (Fig. S6). Further, as shown in Fig. 3B-E and Fig. S7, the AS + DM mice displayed a more severe atherosclerotic status than their AS counterparts.

QC for the metabolomic analysis was conducted using Pearson correlation analysis, ensuring stable and reliable metabolome results. High Pearson correlation coefficients were observed for the serum ESI-, ESI+ and COMB QC models (Fig. S8A-C), reinforcing the credibility of the data. In these modes, a total of 2065, 3739, and 5804 annotated metabolites were identified.

Distinct metabolite clusters in AS + DM mice compared with AS mice were delineated via OPLS-DA score plots for the ESI- (Fig. S9A), ESI+ (Fig. S9B), and COMB (Fig. 3F). Validation model computations of R2 and Q2 confirmed the reliability and absence of overfitting in the OPLS-DA models for ESI- (Fig. S9C), ESI+ (Fig. S9D), and COMB (Fig. 3G). Volcano plots highlighted the levels of the annotated differential metabolites in AS + DM compared with AS in the ESI- (Fig. S9E), ESI+ (Fig. S9F) and COMB (Fig. 3H). An examination of these modes revealed a total of 791, 1287, and 2081 significantly altered (AS + DM vs. AS, $P < 0.05$) metabolites in the ESI-, ESI+ and COMB, respectively. Log2 fold changes of the top 30 significantly changed metabolites in COMB were demonstrated in Fig. 3I. There was a striking overlap (99.83%) in the annotated metabolites between AS and AS + DM, as depicted in Fig. 3J.

## Identification of significantly altered metabolomic patterns in AS + DM mice compared to AS mice

Upon annotation by KEGG, we observed that the metabolomic changes were predominantly enriched in the "Lipid metabolism", "Amino acid metabolism," and "Carbohydrate metabolism" pathways in the ESI- (Fig. S10A), ESI+ (Fig. S10B) and COMB (Fig. S10C). According to HMDB annotation, the top altered functional classes were "Lipids and lipid-like molecules", "Organoheterocyclic compounds" and "Organic acids and derivatives" in ESI-, ESI+ and COMB (Fig. S10D-S10F).

We further investigated the alterations in lipid metabolism using the Lipid Maps database annotation. The category "Fatty acyls" and class "Fatty acid and conjugates" displayed a more pronounced frequency of significantly altered metabolites compared to other categories and classes in the COMB, as depicted in Fig. 3K-M. Fig. 3N illustrates the Log2 fold changes of these notably altered metabolites within the fatty acid and conjugates class. Notably, we identified several saturated fatty acids, such as icosanoic acid, palmitic acid, and nonadecanoic acid, as among the significantly altered metabolites.

## Convergence of metabolomic profiles in human cohort and murine models of diabetes-associated atherosclerosis

In a concerted effort to delineate the metabolomic landscape of diabetes-related atherosclerotic plaque vulnerability with increased accuracy, we devised a model by integrating annotated metabolites from both human and murine samples. Specifically, "non-T2DM" and "T2DM" in humans were analogous to "AS" and "AS + DM" in mice. Evidence from the OPLS-DA analysis confirmed that this model was reliable and not prone to overfitting (Fig. 4A), a conclusion further

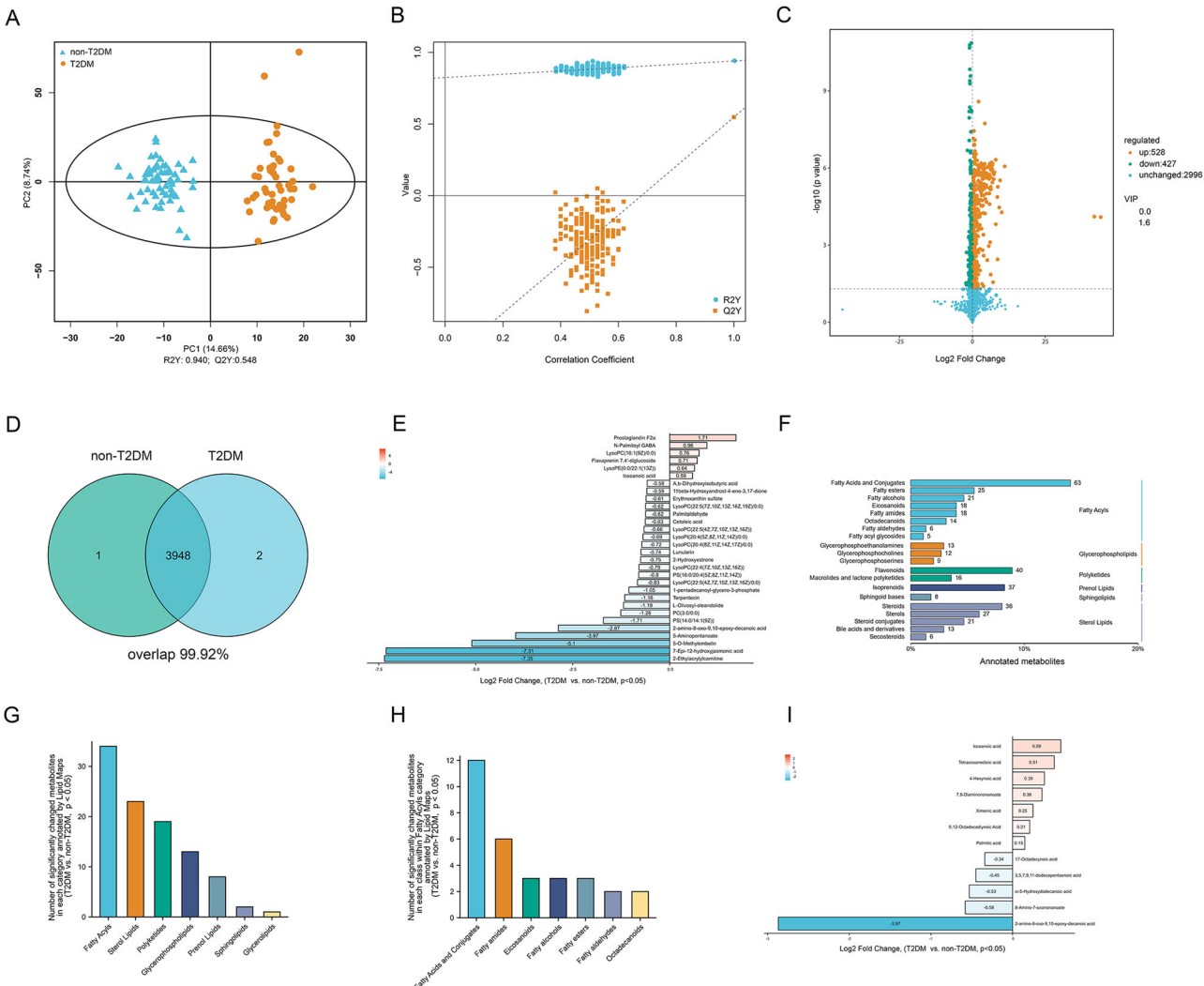

**Fig. 2 | Unveiling divergent metabolomic patterns in T2DM versus non-T2DM subjects within the "Chronic Stable Coronary Atherosclerotic Disease Prospective Cohort" registry. A** Orthogonal partial least squares discriminant analysis (OPLS-DA) scatter plot visualizing metabolomic divergence between non-type 2 diabetes mellitus (non-T2DM) (depicted in blue) and T2DM (depicted in orange) human serum samples in the combined model (COMB). The X and Y axes represent the contributions of individual samples to the first two principal components (PC1 and PC2), respectively. **B** Cross-validation plot of the OPLS-DA model with 200 permutations, indicating robustness and absence of overfitting, as evidenced by intercepts of R2 = (0.0, 0.940) and Q2 = (0.0, 0.548). **C** A volcano plot delineating the pairwise comparisons of metabolites in T2DM relative to non-T2DM subjects. The vertical dashed lines depict the twofold abundance difference threshold, and the horizontal dashed line marks the $P = 0.05$ significance threshold. Metabolites exhibiting significant alterations are highlighted in orange (up-regulated) or green

(down-regulated). **D** Venn diagram illustrating unique and shared metabolites between serum samples from non-T2DM and T2DM subjects in the cohort. **E** Depiction of the top 30 metabolites with the most significant differences based on the absolute values of log2 fold change. The bar graph represents the log2 fold change of each metabolite. **F** Annotation of metabolites using the Lipid Maps database, with text on the left and right sides denoting classes and categories, respectively. **G** Illustration of the frequency of significantly altered metabolites within each category as annotated by Lipid Maps. **H** Representation of the frequency of significantly altered metabolites within each class within the "Fatty Acyls" category. **I** Display of all significantly differentially expressed metabolites within the "fatty acids and conjugates" class based on the significance of their log2 fold change. The bars in the graph represent the log2 fold change values for each metabolite. Source data are provided as a Source Data file.

reinforced by additional validation (Fig. 4B). The volcano plots effectively visualized the magnitude of changes in the annotated differential metabolites common to both humans and mice, when comparing T2DM with non-T2DM (Fig. 4C).

Upon integration, we pinpointed 705 annotated metabolites shared by both humans and mice, as displayed in a venn plot (Fig. 4D). Log2 fold changes of top 30 significantly changed (T2DM vs. non-T2DM, p < 0.05) metabolites in this model were plotted in Fig. 4E. KEGG annotation revealed that the major shifts occurred within the "Lipid metabolism" and "Amino acid metabolism" pathways (Fig. S11A). The HMDB annotation suggested an enrichment of altered metabolites in the functional classes of "Lipids and lipid-like molecules" which is closely tied to lipid metabolism (Fig. S11B). Given these findings, we

utilized the Lipid Maps database for a more thorough exploration of the perturbations in lipid metabolism.

As demonstrated in Fig. 4F, we annotated 34 metabolites within the "Fatty Acyls" category using Lipid Maps database which were shared by humans and mice (Fig. 4H). Log2 fold changes of 17 significantly altered metabolites annotated by Lipid Maps database were further demonstrated in Fig. 4G. Frequencies of altered metabolites within the category "Fatty acyls" and the class "Fatty acid and conjugates" were illustrated in Fig. 4I–K presents the Log2 fold changes of significant metabolites shared between humans and mice, specifically within the "Fatty acyls" category as annotated by Lipid Maps. Notably, among these metabolites, palmitic acid, tetracosanedioic acid, and icosanoic acid—all of which are saturated fatty acids—were observed to

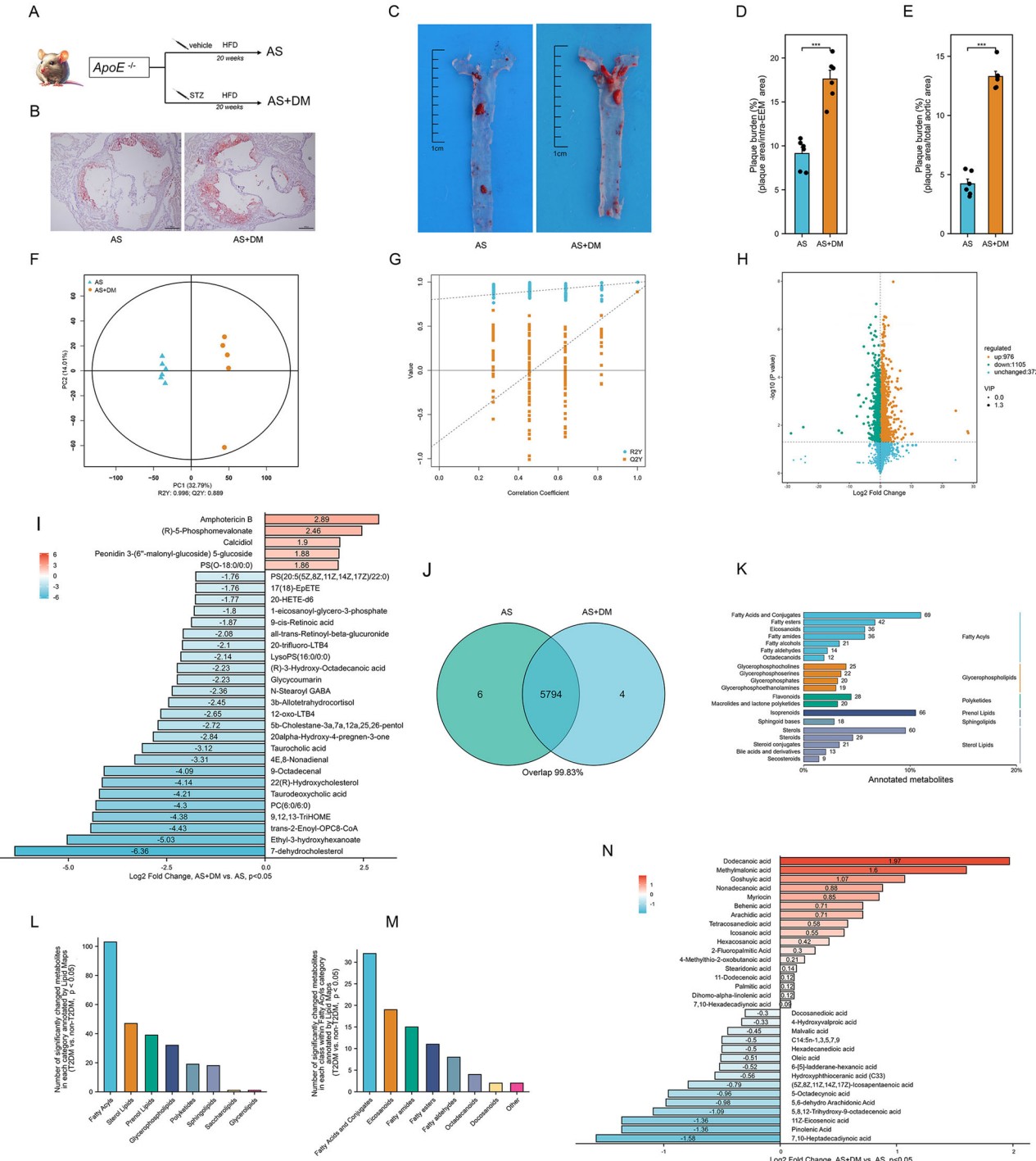

**Fig. 3 | Distinct metabolomic signatures in atherosclerotic mice complicated with diabetes.** 8-week-old male C57BL/9 background *ApoE⁻/⁻* mice and *ApoE⁻/⁻* mice received STZ injections were fed with HFD for 20 weeks. **A** A flowchart of establishing mouse models for atherosclerosis (AS) and atherosclerosis complicated with diabetes (AS + DM). Oil Red O (ORO) staining of aortic root plaques (**B**) and aorta (**C**) in AS and AS + DM mice. Scale bars: 400 μm (**B**), 1 cm (**C**). Quantitation of plaque area in AS and AS + DM mice expressed as the ratio of ORO-positive area to the intra-external elastic membrane (EEM) area (**D**, *n* = 6 mice, data presented as mean ± SD, ***P < 0.001)), and ORO-positive area to total aortic area (**E**, *n* = 6 mice, data presented as mean ± sd, ***P < 0.001). **F** Orthogonal partial least squares discriminant analysis (OPLS-DA) scatter plot illustrating the distinct metabolomic profiles between AS (blue) and AS + DM (orange) mice serum samples in the combined model (COMB). The axes denote the contributions of individual samples to the first two principal components (PC1 and PC2). **G** Cross-validation plot of the OPLS-DA model, corroborated by 200 permutations. The intercepts of

R2 = (0.0, 0.996) and Q2 = (0.0, 0.889) confirm the model's robustness and lack of overfitting. **H** A volcano plot delineating the pairwise comparisons of metabolites in AS + DM relative to AS mice. The vertical dashed lines depict the twofold abundance difference threshold, and the horizontal dashed line marks the *P* = 0.05 significance threshold. Metabolites exhibiting significant alterations are highlighted in orange (up-regulated) or green (down-regulated). **I** Top 30 significantly changed metabolites by log2 fold change. **J** A Venn diagram visualizing unique and shared metabolites of AS + DM and AS mice. **K** Metabolite annotations are performed using the Lipid Maps database, with text on either side signifying classes or categories. Illustration of the frequency of significantly altered metabolites within each category (**L**) and class (**M**) under the "Fatty Acyls" category as annotated by Lipid Maps. **N** Display of all significantly altered metabolites within the "fatty acids and conjugates" class based on the significance of their log2 fold change. Source data are provided as a Source Data file.

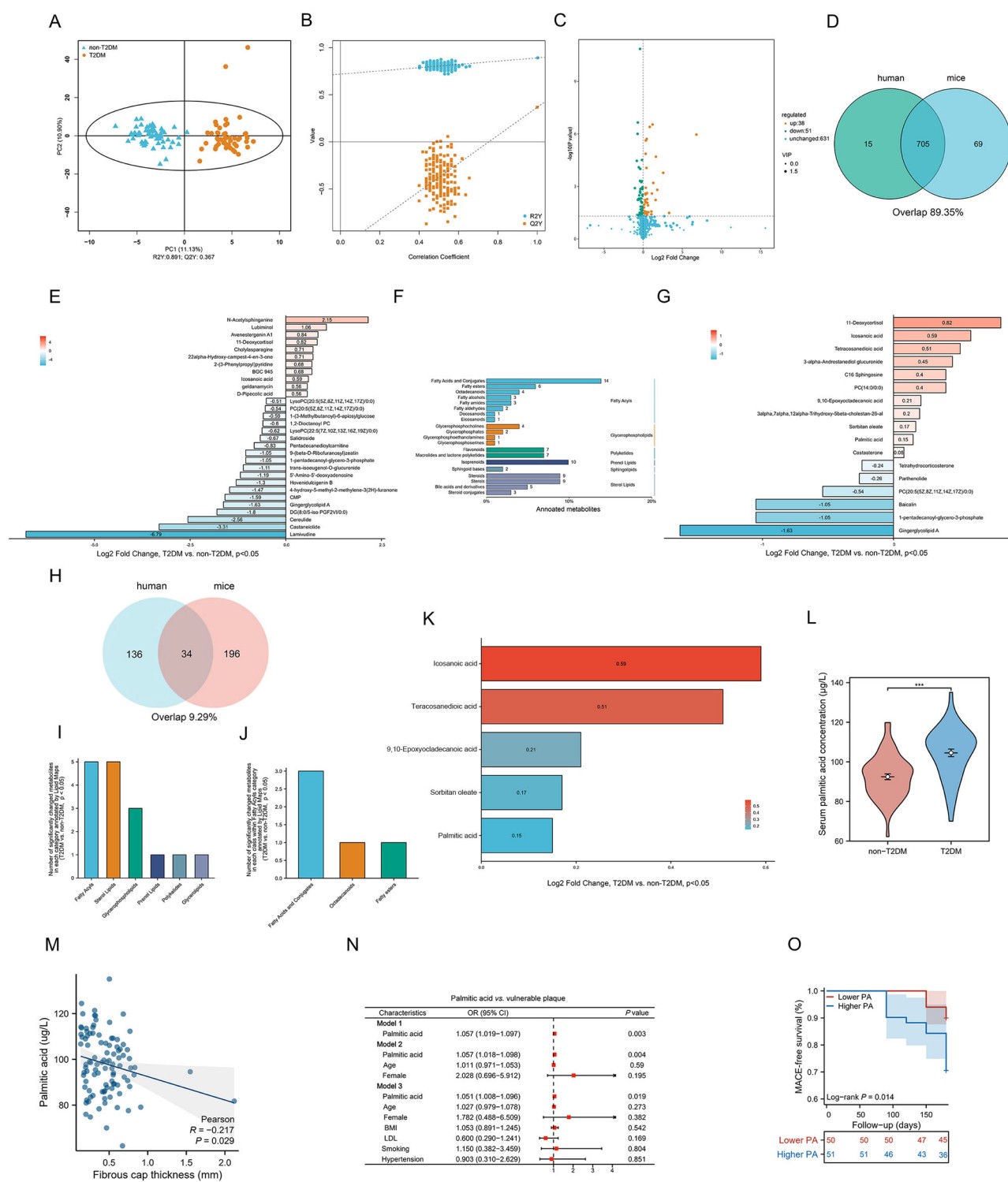

be significantly elevated in T2DM subjects relative to their non-T2DM counterparts.

## Association of elevated palmitic acid levels in T2DM with atherosclerotic plaque vulnerability

Further investigation of the association between significantly changed metabolites identified by this human-mouse convergence model and atherosclerotic plaque vulnerability was conducted. Firstly, by linear correlation analysis, among these significantly changed metabolites, there were 8 metabolites, namely 1-pentadecanoyl-glycero-3-phosphate, 11-Deoxycortisol, cholylasparagine, palmitic acid, icosanoic acid, isolinderenolide, Ornithylamphotericin methyl ester and

tilmicosin, significantly associated with plaque fibrous cap thickness which is a recognized characteristic of vulnerable plaques (Table 3). We stratified the participants in our cohort based on the median serum metabolites concentration, generating higher and lower metabolites level groups. A 180-day follow-up study was carried out to testify weather these metabolites were associated with major adverse cardiovascular events (MACE) which were recognized as the main clinical manifestation of vulnerable atherosclerotic plaques. The MACE included all-cause mortality, ACS, emergency recurrence of percutaneous coronary intervention (PCI), emergency coronary artery bypass grafting and acute cerebral infarction which were highly related to vulnerability of atherosclerotic plaques. As demonstrated in Fig. S12, except

**Fig. 4 | Association between palmitic acid and atherosclerotic plaque vulnerability in T2DM. A** Orthogonal partial least squares discriminant analysis (OPLS-DA) scatter plot demonstrates the metabolomic differences between non-T2DM and T2DM human serum samples. **B** Cross-validation plot for the OPLS-DA model reveals model robustness and absence of overfitting, as validated by the intercepts of R2 = (0.0, 0.891) and Q2 = (0.0, 0.367). **C** Volcano plot detailing the results of pairwise comparisons of metabolites in T2DM relative to non-T2DM subjects. **D** Venn diagram displays the overlap of 705 annotated metabolites shared between human and mouse models. **E** Display of the top 30 altered metabolites, ranked by their log2 fold change. **F** Metabolite annotation using the Lipid Maps database. **G** Visualization of significantly altered metabolites annotated by Lipid Maps database. **H** Venn diagram shows the 34 shared human-mouse annotated metabolites classified within the "Fatty acyls" category according to the Lipid Maps database. Depiction of the count of distinctively altered metabolites across categories (**I**) and within "Fatty acyls" category (**J**). **K** Visualization of significantly altered metabolites in the "Fatty acyls" category. **L** Comparison of serum palmitic acid (PA) concentration between non-T2DM and T2DM human subjects. (n = 55 subjects in non-T2DM and n = 46 subjects in T2DM, data presented as mean ± SD, two-tailed unpaired Student t-test with Welch's correction, ***$P < 0.001$). **M** Two-tailed Pearson's linear regression analysis shows a significant negative correlation (r = −0.262, $P = 0.008$) between serum PA concentration and coronary plaque fibrous cap thickness. The shaded area around the regression line is the error band, representing confidence interval. **N** Logistic regression analysis reveals the relationship between PA and plaque vulnerability in the cohort study. Model 1 indicates the univariate regression analysis; Model 2 is adjusted for age and gender; Model 3 is adjusted for age, gender, BMI, LDL, smoking, and hypertension. (The center for the error bars is the point estimate of the odds ratio for each variable. n = 101 subjects). **O** A Kaplan-Meier survival curve plots the 180-day MACE-free survival in Higher PA group versus Lower PA group. Source data are provided as a Source Data file.

for palmitic acid, no significant correlations were found between MACE and other 7 identified metabolites (all Log-rank $P > 0.05$).

Therefore, among these metabolites, palmitic acid garnered our attention. This interest was not only due to its significant correlation with the IVUS echographic features and clinical manifestations of vulnerable atherosclerotic plaques but also because of its previously documented association with T2DM and coronary artery disease in the literature[15,16]. We observed a significantly higher serum concentration of palmitic acid in T2DM subjects compared to non-T2DM individuals [(104.53 ± 12.97) vs. (92.48 ± 18.87) µg/L, $P < 0.001$] (Fig. 4L). Linear correlation analysis between serum palmitic acid concentrations and IVUS characteristics of coronary atherosclerotic plaques unveiled a significant negative correlation between serum palmitic acid levels and plaque fibrous cap thickness (R = -0.217, $P = 0.029$) (Fig. 4M).

Subsequent regression analysis revealed palmitic acid as a determinant of vulnerable plaque [OR 1.057 (1.019−1.097); $P = 0.003$]. Importantly, the association of palmitic acid with vulnerable plaque [OR 1.051 (1.008−1.096); $P = 0.019$] persisted even after adjusting for potential confounders, such as age, gender, BMI, LDL, smoking, and hypertension (Fig. 4N). The 180-day follow-up study indicated that individuals with elevated serum palmitic acid concentrations experienced a higher incidence of MACE (Log-rank $P = 0.014$) (Fig. 4O). Collectively, these findings highlight the potential role of palmitic acid accumulation in T2DM as a contributor to the vulnerability of atherosclerotic plaques.

## Macrophage-specific Dll4 ablation attenuates palmitic acid-induced notch pathway activation in VSMCs and mitigates atherosclerotic plaque vulnerability

Our previous work indicated that the heightened expression of Dll4 in plaque macrophages induces Notch signaling activation in VSMCs,

thereby exacerbating the atherosclerotic phenotype[10]. In this study, to explore the potential role of macrophage Dll4 in the vulnerability of plaques associated with palmitic acid, we generated macrophage-lineage conditional *Dll4* knock-out atherosclerotic mice [(*Dll4*flox/flox; *Lyz2*-Cre$^{+/-}$; *ApoE*$^{-/-}$) mice, KO] (Fig. S13). Expression of Dll4 was effectively silenced in isolated primary macrophages (Fig. S14G, H), and atherosclerotic plaques (Fig. 5E and Fig. S14H) and isolated aortic macrophages (Fig. S15) of KO.

In our study, we categorized animal subjects into eight groups: WT, WT + HFD, WT + PA, WT + HFD + PA, KO, KO + HFD, KO + PA, and KO + HFD + PA (Fig. S13). Administration of palmitic acid to WT and KO resulted in significantly increased serum total cholesterol (TC) and LDL concentrations (Fig. S14C, E). Our findings revealed that HFD significantly increased plaque burden (Fig. 5A), but it didn't notably alter the plaque's collagen content (Fig. 5B and Fig. 5C), and cell senescence (Fig. 5F) or the expression levels of Dll4 and NICD1 (Fig. 5D, E and Fig. S16). In contrast, palmitic acid, irrespective of HFD, markedly increased plaque burden (Fig. 5A) reduced the plaque's collagen content (Fig. 5B and Fig. 5C), upregulated Dll4 and NICD1 expression (Fig. 5D, E, Fig. S16) and induced cell senescence (Fig. 5F and Fig. S17), emphasizing its role in plaque vulnerability. Comparatively, KO showed reduced plaque burden and increased collagen content, indicating an improvement in plaque vulnerability (Fig. 5A−C). Furthermore, a reduction in Dll4 expression in macrophages (Fig. 5D and Fig. S16A) and NICD1 expression in VSMCs (Fig. 5E and Fig. S16B), as well as cell senescence (Fig. 5F and Fig. S17) were observed in the atherosclerotic plaques of KO compared to WT, both of which received palmitic acid administration.

## *Dll4* silencing attenuates notch signaling-mediated VSMC senescence by suppressing palmitic acid-induced macrophage TLR4 pathway activation

Palmitic acid incubation significantly increased the expression of CD68, indicative of M1 polarization in macrophages (Fig. 6A and Fig. S18). *Dll4* was silenced using small RNA interference, which reduced Dll4 expression in palmitic acid-incubated macrophages (Fig. 6A, Fig. S19E, and Fig. S21D). Palmitic acid incubation upregulated TLR4 expression, ERK phosphorylation, FOXC2 expression, and Dll4 expression in macrophages, suggesting that palmitic acid increased Dll4 expression by activating the TLR4/ERK/FOXC2 pathway in macrophages. Transfection of *Dll4* siRNA effectively silenced Dll4 expression without affecting TLR4/ERK/FOXC2 activation in macrophages (Fig. 6B, Fig. S19A−F, Fig. S21A, and Fig. S21D).

We established contact and non-contact cell co-culture models (Fig. 6C) to investigate whether Notch pathway activation and phenotypic conversion in VSMCs was dependent on macrophages expressing Dll4. Palmitic acid-incubated macrophages, characterized by high Dll4 expression, induced increased synthetic phenotypic marker osteopontin (OPN) expression as well as NICD1 nuclear

## Table 3 | Pearson linear correlation analysis of significantly changed metabolites in human-mouse convergence model and plaque fibrous cap thickness

| Metabolite name | R value | P value* |
|---|---|---|
| 1-pentadecanoyl-glycero-3-phosphate | −0.234 | 0.019 |
| 11-Deoxycortisol | 0.249 | 0.012 |
| Cholylasparagine | 0.219 | 0.028 |
| Palmitic acid | −0.262 | 0.008 |
| Icosanoic acid | −0.21 | 0.035 |
| Isolinderenolide | −0.286 | 0.004 |
| Ornithylamphotericin methyl ester | 0.244 | 0.014 |
| Tilmicosin | 0.313 | 0.01 |

*P values were calculated using the two-tailed standard Pearson correlation coefficient (based on t-distribution, not corrected for multiple comparisons), the correlation coefficient ranges from −1 to 1.

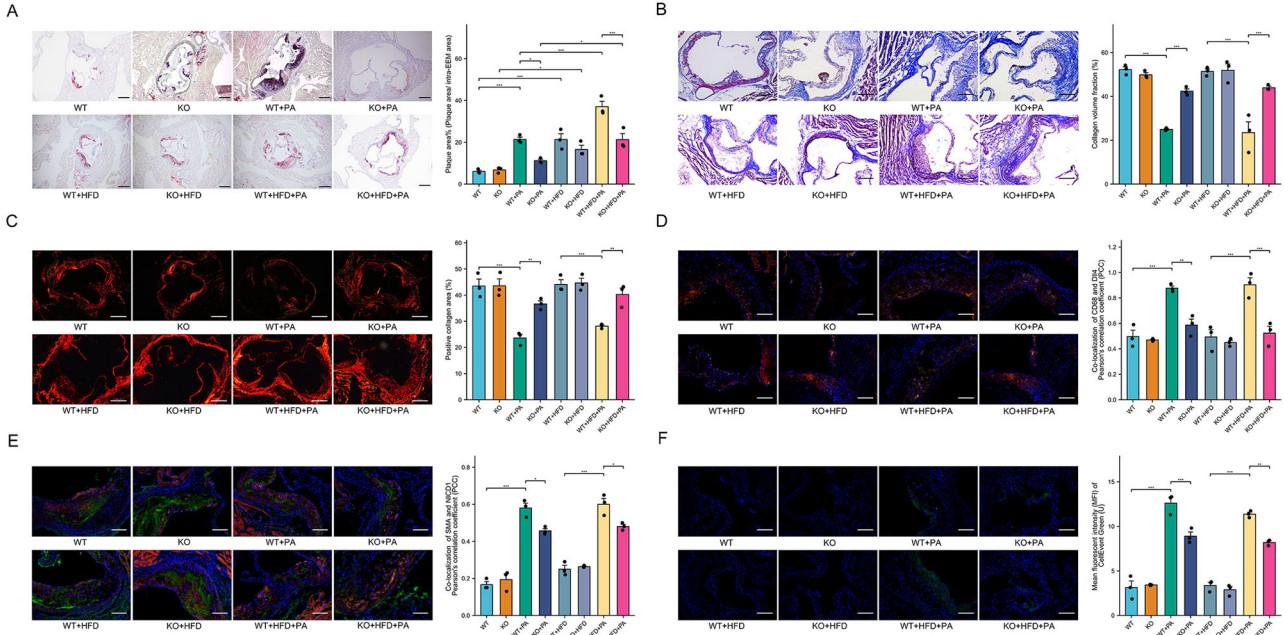

**Fig. 5 | Mitigation of palmitic acid-induced atherosclerotic plaque vulnerability by macrophage Dll4 deletion.** 8-week-old male C57BL/9 background *ApoE^-/-* mice, (*Dll4*^flox/flox^; *Lyz2*-Cre^+/-^; *ApoE^-/-^*) mice were fed with standard chow, high-fat diet (HFD) or HFD containing 5% palmitic acid for 20 weeks. **A** Displayed are exemplary images of aortic root plaques stained with oil red O (ORO). The scale bar indicates a length of 200 μm (*n* = 3 mice, data presented as mean ± SD, left to right: ***$P$ = 0.0007, ***$P$ = 0.0007, *$P$ = 0.0363, *$P$ = 0.0274, ***$P$ = 0.0005, *$P$ = 0.0290, ***$P$ = 0.0005). **B** Depicted are representative images of aortic root plaques subjected to Masson's trichrome staining. The scale bar denotes a length of 200μm (*n* = 3 mice, data presented as mean ± SD, left to right ***$P$ < 0.001, ***$P$ = 0.0008, ***$P$ < 0.001, ***$P$ = 0.0001). **C** Presented are selected images of aortic root plaques stained with Sirius Red. The scale bar signifies a length of 200μm (*n* = 3 mice, data presented as mean ± sd, left to right ***$P$ < 0.001, *$P$ = 0.0373, *** $P$ = 0.0005, **$P$ = 0.0078).

**D** Shown are sample images of aortic root plaques that have undergone double immunofluorescent staining for CD68 and Dll4. The scale bar represents a length of 200μm (*n* = 3 mice, data presented as mean ± sd, left to right ***$P$ = 0.0002, **$P$ = 0.0038, ***$P$ < 0.001, ***$P$ = 0.0002). **E** Exhibited are representative images of aortic root plaques that have undergone double immunofluorescent staining for α-smooth muscle actin (α-SMA) and notch intracellular domain 1 (NICD1). The scale bar equates to a length of 200μm (*n* = 3 mice, data presented as mean ± sd, left to right ***$P$ < 0.001, *$P$ = 0.0168, ***$P$ < 0.001, *$P$ = 0.0208). **F** Depiction of CellEvent Green Senescence staining of the aortic root. The scale bar represents 200μm. (*n* = 3 mice, data presented as mean ± sd, left to right ***$P$ < 0.001, ***$P$ = 0.0004, ***$P$ < 0.001, **$P$ = 0.0021). **A–F**: one-way ANOVA with the Tukey post hoc correction. Source data are provided as a Source Data file.

translocation in VSMCs in the contact cell co-culture model rather than the non-contact model (Fig. 6D and Fig. S21E). Consequently, contact with palmitic acid-incubated macrophages exhibiting high Dll4 expression led to increased expression of HES1, SIRT1, and P21 in VSMCs. However, contact with palmitic acid-incubated macrophages transfected with *Dll4* siRNA suppressed both Notch pathway activation and VSMCs synthetic phenotypic conversion (Fig. 6E, Fig. S20A−F, Figs. S21B, S22).

The activation of the Notch pathway HES1/SIRT1/P21 can lead to senescence[17]. Prior to the assessment of senescence, no significant alteration of pHi was found, ensuring that any potential alterations in pHi would not confound our subsequent senescence analyses (Fig. S23). Macrophages with palmitic acid-induced high Dll4 expression mediated Notch pathway activation in VSMCs, which ultimately displayed a senescent phenotype. Nevertheless, impaired senescence was observed in VSMCs co-cultured with palmitic acid-incubated macrophages transfected with *Dll4* siRNA (Fig. 6F and Fig. S21F).

## Discussion

The intricate relationship between T2DM and atherosclerotic plaque vulnerability has been a focal point of cardiovascular research. Our findings underscore the heightened risk of plaque rupture in T2DM patients, primarily attributed to the thinning of the fibrous cap. While the thinning of this cap is a well-documented precursor to severe cardiovascular events, our study delved deeper into the mechanistic role of palmitic acid in this process. Elevated serum palmitic acid concentrations in T2DM patients, as observed in our study, align with previous research highlighting the dysregulation of lipid metabolism

in diabetes[18]. The consequential increase in palmitic acid levels appears to be a significant factor in the development of vulnerable plaques, as evidenced by our murine model findings. Furthermore, the upregulation of Dll4 expression in macrophages within atherosclerotic plaques and its subsequent influence on VSMC senescence offers a perspective on the cellular interactions that exacerbate plaque vulnerability. By juxtaposing our results with existing literature, we aim to elucidate the multifaceted interplay between T2DM, lipid metabolism, and atherosclerotic plaque dynamics, suggesting potential for targeted therapeutic interventions.

T2DM is characterized by disruptions in lipid metabolism, which have been implicated in the increased vulnerability of atherosclerotic plaques. A key contributor to this vulnerability is the accumulation of advanced glycation end-products (AGEs), which are elevated in T2DM due to persistent hyperglycemia. Interaction of AGEs with their receptor, RAGE, amplifies oxidative stress and inflammation, setting the stage for plaque instability[19]. Furthermore, the dyslipidemia inherent to T2DM augments the presence of oxidized low-density lipoprotein (oxLDL) particles. When macrophages internalize these particles, they transform into foam cells, instigating inflammation and accelerating plaque development[20]. Our findings, derived from both human and murine T2DM models, underscore the association between lipid metabolic disturbances and plaque vulnerability. A salient observation was the pronounced elevation of palmitic acid, a saturated fatty acid (SFA), in the context of T2DM. Established literature confirms that T2DM precipitates lipid metabolic anomalies, manifesting as increased circulating SFAs[21]. This phenomenon is attributed to insulin resistance, which impedes insulin-mediated inhibition of lipolysis in

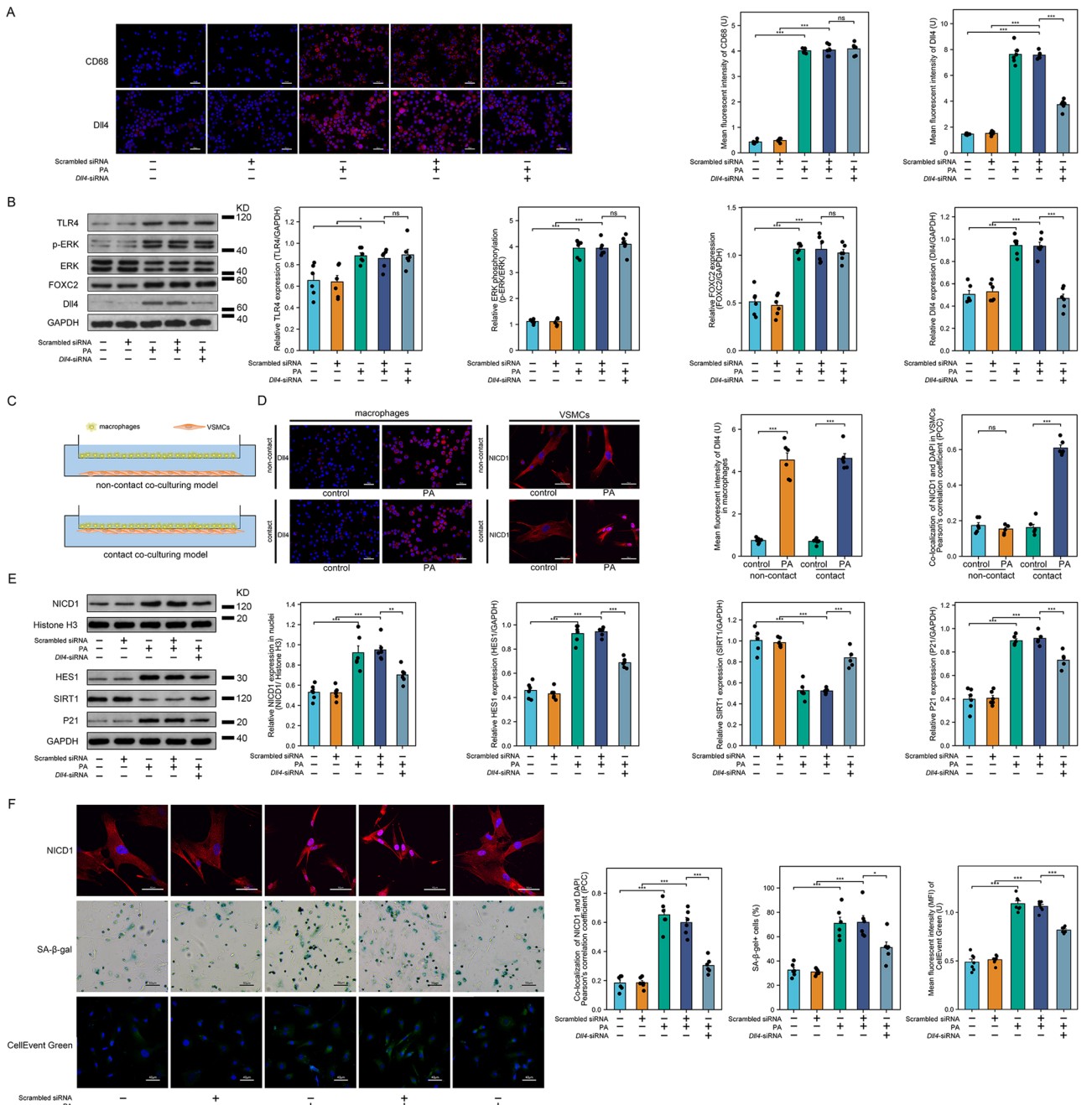

**Fig. 6 | The modulatory role of Dll4 silencing on Notch signaling-mediated VSMC senescence through the suppression of palmitic acid-induced macrophage TLR4 pathway activation. A** Representative images of immunofluorescent staining of CD68 and Dll4 in macrophages, with quantified expression of CD68 (left to right ***$P$ < 0.001, ***$P$ < 0.001, ***$P$ < 0.001) and Dll4 (left to right ***$P$ < 0.001, ***$P$ < 0.001, ***$P$ < 0.001, scale bar = 50 μm). **B** Western blots images and quantifications of relative levels of TLR4 (left to right *$P$ = 0.0202, *$P$ = 0.0298, $P$ = 0.9882), FOXC2 (n left to right ***$P$ < 0.001, ***$P$ < 0.001, ns $P$ = 0.9347), and Dll4 (left to right ***$P$ < 0.001, ***$P$ < 0.001, ***$P$ < 0.001), as well as the relative phosphorylation level of ERK (left to right ***$P$ < 0.001, ***$P$ < 0.001, ns $P$ = 0.8364). **C** A schematic diagram illustrating contact and non-contact co-culturing models of macrophages and VSMCs. **D** Immunofluorescent images of Dll4 in macrophages and NICD1 in VSMCs. Accompanying quantifications illustrate the relative Dll4 expression (left to right ***$P$ < 0.001, ***$P$ < 0.001), and the nuclear translocation of NICD1 (ns $P$ = 0.30945, ***$P$ < 0.001) in VSMCs, scale bar = 50 μm. **E** Western blots images and

quantifications of relative expressions of HES1 (left to right ***$P$ < 0.001, ***$P$ < 0.001, ***$P$ < 0.001), SIRT1 (left to right ***$P$ < 0.001, ***$P$ < 0.001, ***$P$ < 0.001), P21 (left to right ***$P$ < 0.001, ***$P$ < 0.001, ***$P$ = 0.0003), and the nuclear translocation of NICD1 (left to right ***$P$ < 0.001, ***$P$ < 0.001, **$P$ = 0.0032) in VSMCs. **F** Immunofluorescent staining of NICD1 (scale bar = 50μm), SA-β-gal staining (scale bar = 50 μm), and CellEvent Green staining (scale bar = 40 μm) in VSMCs. The corresponding quantifications stand for nuclear translocation of NICD1 by colocalization analysis with DAPI in VSMCs (left to right ***$P$ < 0.001, ***$P$ < 0.001, ***$P$ < 0.001), the extent of VSMC senescence by quantifying the percentage of SA-β-gal positive stained VSMCs (left to right ***$P$ < 0.001, ***$P$ < 0.001, *$P$ = 0.0108) and mean fluorescent intensities of CellEvent Green staining (left to right ***$P$ < 0.001, ***$P$ < 0.001, ***$P$ < 0.001). $n$ = 6 independent replicates in all experiments, All data are presented as mean ± SD. **A**, **B**, **E**, **F**: one-way ANOVA with the Tukey post hoc correction, **D**: Two-tailed unpaired Student t-test and Two-tailed unpaired Student t-test with Welch's correction. Source data are provided as a Source Data file.

adipose tissues, leading to an efflux of SFAs[22]. Compounding this, the liver in T2DM scenarios often hypersecretes very-low-density lipoprotein (VLDL) particles, which are triglyceride-rich and have an enriched SFA profile[23].

Palmitic acid, a prevalent SFA in circulation, has established links to the onset and progression of atherosclerosis and coronary artery disease[24]. Our findings underscore that individuals with T2DM exhibit significantly elevated palmitic acid levels compared to non-T2DM counterparts. Intriguingly, this elevation correlated robustly with heightened atherosclerotic plaque vulnerability. Delving into the mechanistic role of palmitic acid, it emerges as a pivotal regulator of macrophage dynamics, influencing their activation and polarization[25]. Specifically, palmitic acid has been shown to skew macrophages towards the pro-inflammatory M1 phenotype, amplifying inflammation and, consequently, enhancing plaque vulnerability[26]. A salient molecular mechanism underpinning this effect is the activation of the TLR4 receptor by palmitic acid[27]. This receptor activation sets off a cascade of intracellular signaling events, notably involving the extracellular signal-regulated kinase (ERK) and the transcription factor forkhead box C2 (FOXC2)[28]. ERK, a member of the mitogen-activated protein kinase (MAPK) family, participates in regulating multiple cellular functions, including gene transcription, proliferation, and differentiation[29]. Recent insights suggest that the TLR4/ERK/FOXC2 axis culminates in the upregulation of delta-like ligand 4 (Dll4)[28]. In alignment with this, our data revealed an upregulation of Dll4 in macrophages under M1 polarization, driven by palmitic acid via the TLR4/ERK/FOXC2 pathway. Moreover, our data also confirmed that Dll4 expression was up-regulated in atherosclerotic macrophages isolated from diabetic animals (Fig. S24).

In the current study, using both contact and non-contact cell coculture models, our data showed an increased translocation of Notch intracellular domain 1 (NICD1) to the nuclei of VSMCs when in direct contact with macrophages expressing elevated Dll4 levels due to palmitic acid stimulation. The Notch signaling pathway, a conserved cellular communication mechanism, regulates a range of cellular functions, including proliferation and apoptosis[30]. Hairy and enhancer of split-1 (HES1), a transcriptional repressor, is directly modulated by Notch signaling[31]. Studies have shown that HES1 inhibits sirtuin 1 (SIRT1), a NAD-dependent deacetylase, through promoter binding[17]. This inhibition of SIRT1 in VSMCs leads to elevated levels of the cyclin-dependent kinase inhibitor P21, a known regulator of cell cycle arrest and senescence[17]. The clinical implications of this molecular cascade could be foreseen, as the HES1/SIRT1/P21 axis is central to VSMC senescence, a factor in atherosclerotic progression and plaque instability[32]. A consequence of VSMC senescence is reduced collagen synthesis and deposition in the fibrous cap, resulting in its thinning[33]. Thinner fibrous caps are more prone to rupture, increasing the risk of vascular events such as myocardial infarction and stroke. In our analysis of macrophages with Dll4 knockout in atherosclerotic mice models, we observed a reduction in plaque accumulation and an enhancement in collagen deposition, suggesting augmented plaque stability. The lack of Dll4 expression in these macrophages correlated with a diminished NICD1 expression in the VSMCs present in atherosclerotic plaques, indicating potential inhibition of the Notch signaling pathway. Moreover, the targeted silencing of Dll4 in macrophages appeared to attenuate the activation of the TLR4 pathway in response to palmitic acid exposure. This, in turn, led to a subsequent decrease in Notch signaling activity within VSMCs.

To sum up, our research highlights the critical influence of palmitic acid and macrophage Dll4 on atherosclerotic plaque vulnerability in T2DM patients. We've discovered that elevated palmitic acid levels in T2DM patients enhance plaque instability through macrophage activation and subsequent VSMC senescence, with the TLR4/ERK/FOXC2 pathway being a key mediator. Furthermore, our study emphasizes the crucial role of Dll4 in macrophages in this process, as

its knockout attenuates plaque vulnerability. This effect is partly due to the suppression of the palmitic acid-triggered TLR4 pathway and subsequent mitigation of Notch signaling-induced VSMC senescence.

While our study elucidates the intricate interplay between T2DM, palmitic acid levels, and atherosclerotic plaque vulnerability, certain limitations warrant mention. Primarily, the cohort size, although adequately powered for our primary endpoints, might not capture the full spectrum of clinical variability seen in broader populations. The utilization of the *ApoE*[-/-] mouse model, although a standard in cardiovascular research, has distinct physiological differences from humans, necessitating careful interpretation when translating findings to clinical scenarios. An optimal model for our study might combine diabetes with elevated endogenous palmitic acid. However, understanding the sources of this acid, such as potential dysregulation of gut microbiota in diabetes, is crucial. Further exploration in this direction could lead to a more refined animal model for our research. The cross-sectional design of our study inherently restricts the ability to draw robust causal relationships. The association between diabetic cardiovascular complications and alteration of other lipids, such as lysophosphatidylcholine and sphingolipids (Table S5), still needs further in-depth investigation. From a clinical perspective, the data highlight the potential therapeutic implications of modulating palmitic acid-mediated pathways in T2DM patients at elevated cardiovascular risk. The association between increased palmitic acid concentrations and plaque vulnerability posits that interventions targeting these pathways could be pivotal in mitigating cardiovascular events in T2DM. Future research should further elucidate these mechanisms and evaluate the efficacy of potential therapeutic strategies, emphasizing their translational potential and impact on patient outcomes.

## Methods

### Ethics statement
All procedures and protocols involving human participants in this study were conducted following the principles of the Declaration of Helsinki and were approved by Shaanxi Provincial People's Hospital Institutional Review Board. Informed written consent was obtained from all individuals before their inclusion in the study. The privacy and confidentiality of all participants were rigorously maintained. In terms of animal use, all experimental procedures were performed according to the guidelines outlined by Shaanxi Provincial People's Hospital Institutional Review Board. The study protocol was reviewed and approved by the same committee. Every effort was made to minimize suffering and reduce the number of animals used. Euthanasia, when necessary, was performed in a humane manner according to the recommendations of the American Veterinary Medical Association (AVMA). All data analyses were performed with strict adherence to ethical guidelines to ensure integrity and prevent any form of data manipulation or falsification. The results were reported honestly and transparently, acknowledging all relevant limitations.

### National inpatient sample (NIS) database study
This investigation utilizes data from the NIS database, the most comprehensive inpatient database publicly accessible in the United States, established by the Agency for Healthcare Research and Quality. The focus of this study is the period spanning from 2016 to 2018.

To distinguish the patterns of hospitalizations, patients were identified and categorized based on a primary diagnosis of either T2DM or non-T2DM. This stratification was facilitated by the International Classification of Diseases, Tenth Revision, and Clinical Modification (ICD-10-CM) codes. An extensive dataset was compiled, incorporating 2,318,638 hospitalizations diagnosed with T2DM and 5,357,761 hospitalizations identified as non-T2DM (Fig. 1A). To ensure data integrity and relevance, we excluded cases involving patients under 18 years of age or those with incomplete data.

 

Beyond the primary diagnosis, the study sought to assimilate additional information pertaining to patients' demographic characterstics and cardiovascular risk profiles, including a history of coronary artery disease (CAD), smoking history, dyslipidemia, and hypertension. Of paramount importance was the exploration of diseases highly associated with vulnerable atherosclerotic plaques, such as acute coronary syndrome (ACS, including acute myocardial infarction or unstable angina pectoris) and cerebral infarction, supported by corresponding ICD-10-CM codes (Table S1). The study aimed to elucidate trends related to the (1) prevalence of comorbidities, (2) inpatient mortality rates, (3) average length of hospital stay, and (4) mean cost associated with these stays.

### Prospective cohort registry study

**Study population.** The present investigation was anchored within the "Chronic Stable Coronary Atherosclerotic Disease Prospective Cohort" registry (ClinicalTrials.gov identifier: NCT05270330). This ongoing prospective study explores the evolution of endovascular imaging. The cohort was assembled from July 1, 2021, to July 31, 2022.

Inclusion criteria admitted patients aged 18 to 85 years, presenting with stable angina pectoris, and having undergone invasive diagnostic cardiac angiography and IVUS evaluation. The applied definition of stable angina pectoris was chest pain upon exertion that showed no change in frequency, intensity, or duration over the preceding four weeks, which is in accordance with current guidelines[34]. Participants were from the same geographical region, ensuring consistent dietary habits. None had prior diagnoses of coronary heart disease or T2DM, and thus were not on related medications. Additionally, all had uniform physical activity levels without specific restrictions. Exclusion criteria included a history of coronary artery bypass graft (CABG) or PCI, and the presence of severe chronic kidney disease, cardiogenic shock, valvular heart disease, heart failure, blood disorders, thyroid disease, severe hepatic or renal dysfunction, recent wounds, ongoing chronic inflammatory, autoimmune diseases or malignant conditions. A total of 101 patients were enrolled in our study, as detailed in Fig. S1. The case report form (CRF) used in this study is presented in Table S2.

**Coronary angiography procedure and analysis.** Coronary angiography analyses were performed using an Artis zee system (Siemens Healthineers). Prior to image acquisition, 200 μg of intracoronary nitroglycerin was administered to ensure maximal arterial dilation. The procedure involved percutaneous access, typically through the radial or femoral artery, introduction of a guide catheter into the ostium of the coronary artery, and injection of a contrast agent. Quantitative measurements were taken from digital recordings of the angiograms, analyzed offline by blinded operators at the cardiovascular core laboratory using CAAS software (Version 5.10, Pie Medical Imaging).

**IVUS image acquisition and analysis.** IVUS images were procured utilizing the iLab™ POLARIS Multi-Modality Guidance System (Version 2.8.1, Boston Scientific) and a 40 MHz, 2.9 F monorail IVUS catheter (OptiCross™ X Imaging Catheter, Boston Scientific). Following the administration of 200 μg of intracoronary nitroglycerin, the catheter was advanced distal to the lesion of interest. An automated pullback at a speed of 0.5 mm/sec was performed for a comprehensive evaluation of the target vessel. Offline image analysis was performed at a core laboratory using the iReview software (Version 2.0, Boston Scientific). Plaques with intact structural integrity and high lipid content in the culprit vessel were identified. Subsequently, three representative frames were selected for quantitative analysis, and average values were calculated. The cross-sectional area of the external elastic membrane (EEM), lumen area, area occupied by the lipid pool, and thickness of the fibrous cap were measured. The total plaque area was derived by subtracting the lumen area from the EEM area. Plaque burden was assessed using the formula: (EEM area at minimal lumen area site - minimal lumen area) / EEM area at minimal lumen area site. Plaques with a plaque burden greater than or equal to the median value of 80.26% were categorized as "greater plaque burden". These plaques are considered "vulnerable" due to their strong association with acute coronary syndrome (ACS)[35].

**Follow-up and endpoints.** Patients were followed up at baseline and 6 months post-discharge. Follow-up was conducted via telephone calls, regular outpatient follow-up, or electronic hospital records. The primary endpoint of the study was all-cause mortality. Secondary endpoints included recurrence of ACS, emergency recurrence of PCI, emergency coronary artery bypass grafting and acute cerebral infarction. The study maintained a 100% follow-up rate, with no patients lost to follow-up.

### Animal study

**Generation of Dll4 macrophage-conditional knock-out mice with ApoE deficiency.** *ApoE*-deficient mice (B6/JGpt-*Apoe*em1Cd82/Gpt, Strain NO.T001458, *ApoE*-/- mice), mice with floxed *Dll4* alleles (B6/JGpt-*Dll4*em1Cflox/Gpt, Strain NO. T009877, *Dll4*flox/flox mice), and *Lyz2*-Cre mice (B6/JGpt-*Lyz2*em1Cin(CreERT2)/Gpt, Strain NO.T052789) were acquired from GemPharmatech. The mice were maintained in a C57BL/6 strain background and housed in independent cages under controlled environmental conditions with a 12-hour light-dark cycle, at 25°C, and 50% humidity. All mice were provided ad libitum access to water and standard chow (LAD2001, Trophic Animal Feed High-tech Co.).

*Dll4*flox/flox mice were crossbred with *Lyz2*-Cre mice to generate heterozygous (*Dll4*flox/wt; *Lyz2*-Cre+/-)mice. These heterozygous mice were subsequently crossbred with *Dll4*flox/flox mice to yield (*Dll4*flox/flox; *Lyz2*-Cre+/-)mice. To generate *ApoE*-deficient mice with the macrophage-specific *Dll4* knockout, (*Dll4*flox/flox; *Lyz2*-Cre+/-) mice were bred with *ApoE*-/- mice to yield (*Dll4*flox/+; *Lyz2*-Cre+/-; *ApoE*-/-)mice. Further breeding resulted in the generation of (*Dll4*flox/flox; *Lyz2*-Cre+/-; *ApoE*-/-) mice.

**Establishment of animal models and treatments.** At 8 weeks of age, *ApoE*-/- and (*Dll4*flox/flox; *Lyz2*-Cre+/-; *ApoE*-/-) mice were administered intraperitoneal injections of streptozotocin (STZ, Sigma-Aldrich) at a dosage of 50 mg/kg/day for five consecutive days. Mice with blood glucose levels exceeding 300 mg/dL were classified as diabetic. Subsequently, the mice were fed with standard chow or high-fat diet (HFD, D12079B, Research Diets Inc.) for 20 weeks. This HFD derived 42% of its caloric content from fat (21 g/100 g diet), 43% from carbohydrates, and 15% from protein, with an additional 0.15% cholesterol content. A one-week acclimatization period preceded the commencement of the dietary intervention. Mice that received palmitic acid treatment were fed with a HFD containing 5% palmitic acid[36] (HFD supplemented with palmitic acid, 5% w/w, customized by Tropic Animal Feed High-Tech Co.) for 20 weeks. In our study, animal euthanasia was conducted using carbon dioxide ($CO_2$) inhalation, in accordance with AVMA guidelines. Blood and fresh tissue were sampled immediately after the animals were sacrificed.

### Histology evaluations

**Oil Red O, Masson's, and Sirius Red stains.** The extracted aortic roots from the mice specimens were processed for histological examination. The specimens were first fixed in 10% neutral buffered formalin solution (Sigma-Aldrich) and subsequently embedded in optimal cutting temperature (OCT) compound (Sakura Finetek). The lipid accumulation within the aortic root sections, cut at 10 μm thickness with a cryostat, was visualized using Oil Red O stain (Sigma-Aldrich). Hematoxylin (Sigma-Aldrich) served as a counterstain to highlight the nuclei.

To assess collagen content, the aortic root sections underwent Masson's Trichrome and Sirius Red staining protocols, respectively. In Masson's Trichrome staining, the sections were sequentially treated

with Weigert's iron hematoxylin (Sigma-Aldrich), Biebrich scarlet-acid fuchsin (Sigma-Aldrich), phosphomolybdic-phosphotungstic acid (Sigma-Aldrich), and aniline blue (Sigma-Aldrich). Sirius Red staining involved an incubation period of one hour in Sirius Red solution (Direct Red 80 in saturated picric acid, Sigma-Aldrich) followed by a wash in acidified water. Post-staining, the sections were dehydrated, cleared, and affixed with a mounting medium (Sigma-Aldrich). The stained sections were subsequently observed under a light microscope and images were captured for further quantitative evaluation.

**Immunofluorescent stains.** To investigate the localization and expression of specific proteins in the aortic root sections, double immunofluorescent staining was performed. The sections were prepared by cutting 10 μm thick slices of aortic root specimens embedded in optimal cutting temperature (OCT) compound (Sakura Finetek). The sections were then permeabilized with 0.3% Triton X-100 (Sigma-Aldrich) in phosphate-buffered saline (PBS) for 15 minutes and blocked with 5% bovine serum albumin (BSA, Sigma-Aldrich) in PBS for 1 hour at room temperature.

For the CD68 and Dll4 staining, primary antibodies anti-CD68 (Cell Signaling Technology, Cat#29176SF, Clone: E3O7V, 1:200 dilution) and anti-Dll4 (Cell Signaling Technology, Cat#96406 S, Clone: D7N3H, 1:200 dilution) were applied to the sections and incubated overnight at 4 °C. The sections were then washed and incubated with Alexa Fluor 488-conjugated anti-rat IgG (1:500, Invitrogen) and Alexa Fluor 594-conjugated anti-rabbit IgG (Jackson ImmunoResearch, Cat# 611585215, 1:500 dilution) for 1 hour at room temperature. For the SMA and NICD1 staining, primary antibodies anti-SMA (1:200, Abcam) and anti-NICD1 (Cell Signaling Technology, Cat#3608 S, Clone: D1E11, 1:200 dilution) were applied to the sections and incubated overnight at 4 °C. Subsequently, the sections were washed and incubated with Alexa Fluor 488-conjugated anti-mouse IgG (1:500, Invitrogen) and Alexa Fluor 594-conjugated anti-rabbit IgG (Jackson ImmunoResearch, Cat#611585215, 1:500 dilution) for 1 hour at room temperature.

After washing, the sections were mounted with Vectashield Antifade Mounting Medium containing DAPI (Vector Laboratories) to counterstain the nuclei. Fluorescent images were acquired using a confocal microscope (Leica Microsystems) and analyzed for colocalization and expression patterns of the targeted proteins.

## Metabolomics profiling

Blood samples designated for metabolomics analysis were gathered from participants within a 12-hour window following invasive coronary evaluation and from mice immediately post-sacrifice, maintaining uniformity in sample collection.

The analytical procedure incorporated liquid chromatography/mass spectrometry (LC/MS) to examine the metabolic extracts obtained through methanol-assisted protein precipitation. The analysis was facilitated by the Acquity I-Class PLUS ultra-high-performance liquid spectrometer (Waters) and the Xevo G2-XS QTOF high-resolution mass spectrometer (Waters), equipped with an Acquity UPLC HSS T3 column (Waters). The mobile phase was composed of a 0.1% formic acid aqueous solution and 0.1% formic acid acetonitrile. An injection volume of 1 μL was consistently utilized.

The Xevo G2-XS QTOF high-resolution mass spectrometer was conFig.d to collect both primary and secondary mass spectrometry data in MSe mode, under the supervision of the acquisition software (MassLynx V4.2, Waters). The system allowed for simultaneous low and high collision energy data acquisition during each cycle, with the low collision energy set at 2 V and the high collision energy ranging between 10-40 V. The scan frequency was established at 0.2 seconds per mass spectrum. ESI ion source parameters were adjusted to the following: capillary voltage, 2000V (positive ion mode) or -1500V (negative ion mode); cone voltage, 30 V; ion source temperature,

150 °C; desolation gas temperature, 500 °C; backflush gas flow rate, 50 L/h; and desolation gas flow rate, 800 L/h.

MassLynx software (V4.2, Waters) was used to capture raw data, which was then processed by Progenesis QI software (V2.0, Waters) for peak extraction, peak alignment, and other data operations. Compound identification was aided by the Progenesis QI software online METLIN database and a self-constructed library (Biomark), ensuring that the theoretical fragment identification and mass deviation were within the defined 100ppm limit. Cut-off values for human and animal metabolomics evaluation were set as 42 and 53 respectively based on average noise intensity.

## Cell study

**Primary cells isolation.** Primary peritoneal macrophages were harvested from mice via peritoneal lavage. Briefly, mice were injected intraperitoneally with 3 ml of sterile 3% Brewer thioglycolate medium (Sigma-Aldrich). Four days post-injection, mice were euthanized, and 8 ml of ice-cold phosphate-buffered saline (PBS, Gibco) was injected into the peritoneal cavity. The peritoneal exudates were collected and centrifuged at 300 x g for 5 minutes. The pellet was resuspended in Dulbecco's Modified Eagle Medium (DMEM, Gibco) supplemented with 10% fetal bovine serum (FBS, Gibco), 100 U/ml penicillin and 100 μg/ml streptomycin (Gibco). The cells were seeded in tissue culture plates and allowed to adhere for 2 hours at 37 °C with 5% $CO_2$, after which the non-adherent cells were removed, and the adherent macrophages were used for further experiments.

Aortic vascular smooth muscle cells (VSMCs), macrophages, and endothelial cells were isolated from mice aortas[37–39]. Briefly, aortas were carefully dissected free of connective tissue and opened longitudinally. The intimal layer was gently scraped off to enrich for smooth muscle cells. The tissue was then minced and digested in a solution containing 1 mg/ml collagenase type II (Worthington Biochemical Corporation) and 0.5 mg/ml elastase (Worthington Biochemical Corporation) at 37 °C for 1 hour. Following digestion, the cells were dispersed in DMEM supplemented with 20% FBS, 100 U/ml penicillin and 100 μg/ml streptomycin. The cell suspension was filtered through a 70 μm cell strainer (BD Biosciences) to remove undigested tissue and debris, and the filtrate was centrifuged at 300xg for 5 minutes. The cell pellet was resuspended and cultured in DMEM supplemented with 20% FBS and antibiotics. The yield VSMCs were allowed to proliferate until they reached 80% confluence. The endothelial cells are then dislodged using a cell scraper, and the resulting cell suspension was centrifuged at 300xg for 5 minutes. The cell pellet is resuspended in Endothelial Cell Growth Medium (EGM-2, Lonza) supplemented with the EGM-2 BulletKit (Lonza). The yield cells were prepared as primary aortic endothelial cells. Approximately 1 g of the aortic tissue enriched of atherosclerotic plaques was finely minced and subjected to a thorough wash in Hank's Balanced Salt Solution with reduced calcium concentration (0.2 mmol/L). Tissue fragments were incubated in 10 mL of HBSS supplemented with 450 units of purified collagenase, 4.7 units of purified elastase, and 1 mg/mL soybean trypsin inhibitor at 37 °C for 2 hours. Post-incubation, the digested tissue was filtered through a 100 μm nylon mesh to separate the cells from undigested tissue fragments. The same tissue was subjected to a second round of enzymatic digestion under identical conditions. The liberated cells were centrifuged at 800xg for 5 minutes, and the cell pellet was resuspended in the supplemented HBSS. A discontinuous density gradient was prepared using metrizamide, with a "light" layer of low-calcium HBSS and a "heavy" layer of 35% metrizamide in low-calcium HBSS, underlain by 1 mL of 38% metrizamide. The cell suspension was carefully layered atop this gradient and centrifuged at 3,000 x g for 20 minutes at 10 °C. The isolated cells were harvested, rinsed, and cultured

overnight in Bio-MPM-1 medium (Biological Industries) supplemented with 15% fetal calf serum (FCS, Fisher). The yield cells were prepared as aortic plaque macrophages.

**Dll4 silencing.** Dll4 expression was silenced in isolated primary macrophages through the introduction of siRNA plasmids designed to target the *Dll4* gene. The transfection process was carried out using Lipofectamine 3000 (Thermo Fisher Scientific), according to the manufacturer's guidelines. Briefly, macrophages were seeded in 6-well plates and grown to 70-80% confluence. The Dll4-targeting siRNA plasmids (Origene, Cat#TL503037) were diluted in Opti-MEM I Reduced Serum Media (Thermo Fisher Scientific) and combined with Lipofectamine 3000 reagent. The siRNA-Lipofectamine 3000 complexes were allowed to form for 20 minutes at room temperature, and then added to the macrophages. After 48 hours of incubation, successful knockdown of Dll4 was confirmed by quantitative PCR and Western blot analysis. Non-targeting scrambled siRNA plasmids (Thermo Fisher Scientific) served as a control in all experiments.

**Cell treatments and apoptosis assay.** Primary macrophages were isolated and subsequently cultured in DMEM supplemented with 10% FBS and 1% penicillin-streptomycin, under a humidified atmosphere containing 5% $CO_2$ at 37 °C. For the purpose of inducing M1 polarization, macrophages were treated with palmitic acid (Sigma-Aldrich). A concentration gradient was established, with cells being exposed to 0, 0.2, 0.4, 0.8, and 1 mmol/L of palmitic acid. Each concentration was maintained for a 6-hour incubation period. Post-treatment, the extent of apoptosis in the macrophages was determined using flow cytometry. This was achieved by staining the cells with an Annexin V-FITC/propidium iodide (PI) apoptosis detection kit (BD Biosciences), strictly adhering to the manufacturer's protocol. Our analysis, as visualized in Fig. S2, indicated that palmitic acid concentrations exceeding 0.4 mmol/L led to pronounced apoptosis in the macrophages. Given this observation, a concentration of 0.4 mmol/L palmitic acid was deemed optimal for subsequent experiments, ensuring the effective induction of M1 polarization without excessive apoptosis.

**Co-culturing.** Two co-culture models, contact and non-contact, were employed in this study[11] (Fig. 6C). Macrophages were co-cultured with VSMCs using a Millicell-24 Cell Culture Insert Plate system equipped with polyethylene terephthalate membranes (Millipore). In the contact co-culture model, macrophages were first seeded onto the basal surface of a cell culture insert at a density of $4 \times 10^4$ cells/cm$^2$. Using RPMI-1640 medium (Gibco) supplemented with 10% fetal bovine serum (FBS, Gibco), 2 mmol/L glutamine (Gibco), and a combination of penicillin and streptomycin (Gibco), the macrophages were allowed to adhere overnight at 37 °C in a humidified atmosphere containing 5% $CO_2$. Following this, the insert was carefully inverted, and VSMCs were seeded onto its apical surface at a density of $4 \times 10^4$ cells/cm$^2$ allowing them to adhere overnight under the same conditions. After the insert was carefully re-inverted, the entire assembly was then incubated for 72 hours in the supplemented RPMI-1640 medium. In contrast, for the non-contact co-culture model, VSMCs were seeded directly onto the bottom of the culture well, ensuring that while both cell types were immersed in the same medium, direct physical interaction was prevented. The rest of the procedure mirrored the contact model, with cells maintained in the supplemented RPMI-1640 medium for the designated period.

**Immunofluorescent stains.** The isolated primary macrophages and vascular smooth muscle cells (VSMCs) were subjected to immunofluorescent staining. Initially, cells were fixed and permeabilized using appropriate reagents. Cells were blocked with blocking buffer

(Abcam). Subsequently, macrophages were stained using anti-CD68 antibody (Cell Signaling Technology, Cat#29176SF, Clone: E3O7V, 1:200 dilution) or anti-Dll4 antibody (Cell Signaling Technology, Cat#96406 S, Clone: D7N3H, 1:200 dilution). VSMCs were treated with an anti-NICD1 antibody (Cell Signaling Technology, Cat#3608 S, Clone: D1E11, 1:200 dilution) or anti-OPN antibody (Cell Signaling Technology, Cat#27927 S, 1:200). Primary antibodies were applied and allowed to incubate overnight at 4 °C. Post-primary antibody incubation, cells were washed by PBS and then incubated with secondary antibodies conjugated with Alexa Fluor 594 (Jackson ImmunoResearch, Cat#611585215, Clone: 611585215, 1:1000 dilution) for 1 hour at room temperature at dark. Cell nuclei were counterstained with DAPI (1:5000, Sigma-Aldrich). The stained cells were then visualized and imaged using a fluorescence microscope. Quantification of fluorescence intensity was performed using ImageJ software (Version 1.53t, NIH).

**Assessment of VSMCs senescence.** To delineate the senescence status of isolated VSMCs, a combination of senescence-associated β-galactosidase (SA-β-gal) staining and CellEvent Senescence Green probe was employed[40]. Notably, the intracellular pH is crucial for SA-β-gal staining, as the enzyme exhibits optimal activity at a pH of 6.0, a hallmark of senescent cells. To ensure that potential variations in intracellular pH (pHi) did not influence SA-β-gal activity, pHi was assessed prior to senescence detection. The Intracellular pH Calibration Buffer Kit (ThermoFisher) was employed to establish a standard curve by clamping intracellular pH values at 4.5, 5.5, 6.5, and 7.5. Subsequently, pHi was ascertained by quantifying cellular fluorescence post-staining with the pHrodo Red Intracellular pH Indicator (ThermoFisher) using a spectrophotometer (Synergy H1, Biotek). The SA-β-gal staining was executed using the Senescence β-Galactosidase Staining Kit (Cell Signaling Technology). For senescence evaluations, VSMCs were first washed with PBS and then fixed using the fixation solution from the kit for 15 minutes at room temperature. Post-fixation, cells were rinsed with PBS and subsequently incubated with the SA-β-gal staining solution (1 mg/mL X-gal, 40 mmol/L citric acid/sodium phosphate, pH 6.0, 5 mmol/L potassium ferrocyanide, 5 mmol/L potassium ferricyanide, 150 mmol/L NaCl, and 2 mmol/L MgCl$_2$) at 37 °C, devoid of $CO_2$, for 16 hours. Under bright-field microscopy, SA-β-gal-positive cells manifested a distinct blue coloration. The percentage of SA-β-gal-positive cells was determined using ImageJ software (NIH), by comparing the blue-stained cell count to the total cell count across random microscopic fields. For further validation, the CellEvent[TM] Senescence Green Detection Kit (ThermoFisher) was applied. After fixation, VSMCs were exposed to the kit's working solution for 2 hours at 37 °C, in the absence of $CO_2$, within a dark environment. After PBS washing, nuclei were stained with DAPI. The resultant stained cells were visualized under a fluorescence microscope, and the fluorescence intensity was quantified using ImageJ software (NIH).

**Quantitative real-time PCR (qRT-PCR)**
Total RNA was extracted from cultured macrophages and VSMCs using the RNeasy Mini Kit (Qiagen) following the manufacturer's instructions. The quantity and purity of the isolated RNA were determined using a NanoDrop spectrophotometer (Thermo Fisher Scientific). Subsequently, 1 μg of total RNA was reverse transcribed into cDNA using the High-Capacity cDNA Reverse Transcription Kit (Applied Biosystems). Quantitative real-time PCR was carried out using the PowerUp SYBR Green Master Mix (Applied Biosystems) on a QuantStudio 5 Real-Time PCR System (Thermo Fisher Scientific). The primer sequences for *TLR4*, *FOXC2*, *Dll4*, *HES1*, *SIRT1*, and *P21*, as well as the reference gene *GAPDH*, were listed in Table S3. The cycling conditions consisted of an initial denaturation at 95 °C for 10 minutes, followed by 40 cycles of denaturation at 95 °C for 15 seconds and annealing/

extension at 60 °C for 1 minute. The relative expression levels of target genes were calculated using the $2^{(-\Delta\Delta Ct)}$ method, with *GAPDH* serving as the reference gene for normalization.

## Western Blots

Cells were lysed employing RIPA buffer supplemented with a protease and phosphatase inhibitor cocktail (all from Sigma-Aldrich). The BCA Protein Assay kit (Pierce Biotechnology) was employed to measure protein concentrations. Equal quantities of protein were resolved on SDS-PAGE, subsequently transferred onto PVDF membranes (Millipore), and probed using specific antibodies. Macrophages were probed with antibodies against TLR4 (Cell Signaling Technology, Cat#14358 S, Clone: D8L5W, 1:1000 dilution), FOXC2 (Cell Signaling Technology, Cat#12974 S, Clone: 04D4, 1:1000 dilution), Dll4 (Cell Signaling Technology, Cat#96406 S, Clone: 96406 S, 1:1000 dilution), phosphorylated ERK (Cell Signaling Technology, Cat#4370 S, Clone: D13.14.4E, 1:1000 dilution) and ERK (Cell Signaling Technology, Cat#4695 S, Clone: 137F5, 1:1000 dilution). VSMCs were probed with antibodies against HES1 (Cell Signaling Technology, Cat#11988 S, Clone: D6P2U, 1:1000 dilution), SIRT1 (Cell Signaling Technology, Cat#9475 S, Clone: D1D7, 1:1000 dilution), P21 (Abcam, Cat#ab109520, Clone: EPR362, 1:1000 dilution). Following primary antibody incubation, membranes were then incubated with horseradish peroxidase-conjugated secondary antibodies (Jackson ImmunoResearch, Cat#111035045, 1:5000 dilution) and visualized using the ECL Western Blotting Substrate (Pierce Biotechnology). Band intensities were quantified using ImageJ software (NIH). GAPDH (Sigma-Aldrich, Cat#G9545, Clone: NA.41, 1:5000 dilution) was probed as a loading control. For the evaluation of nuclear NICD1 expression, nuclear protein was extracted using the NE-PER Nuclear and Cytoplasmic Extraction Reagents (Thermo Fisher Scientific) and probed with anti-NICD1 antibody (Cell Signaling Technology, Cat#3608 S, Clone: D1E11, 1:1000 dilution). Histone H3 (Cell Signaling Technology, Cat#4499 S, Clone: D1H2, 1:1000 dilution) was used as a loading control for nuclear protein.

## Statistics

Statistical analyses were conducted using SPSS Statistics software (version 19.0, IBM). Data are presented as mean ± standard deviation (SD) for normally distributed variables or median with interquartile range for non-normally distributed variables. The Shapiro-Wilk test assessed data distribution normality. For pairwise comparisons, the Student's t-test (two-tailed) was applied to normally distributed data, while the Mann-Whitney U test was utilized for non-normally distributed data. For multiple group comparisons, one-way ANOVA followed by Tukey's post-hoc test was employed for normally distributed data, and the Kruskal-Wallis test followed by Dunn's post-hoc test was used for non-normally distributed data. Pearson's or Spearman's correlation coefficients determined correlations, contingent on data distribution. The associations among T2DM and the ACS, cerebral infarction, and plaque instability were examined via a logistic regression model. Cumulative survival curve for endpoints was constructed using the Kaplan-Meier method and the log-rank test was used to find the difference. Metabolomics data underwent processing and analysis using the XCMS package in R software (version 4.0.3). Preprocessing involved peak detection, retention time alignment, and peak area integration. Metabolite features were identified by matching accurate mass and retention time with an in-house library and public databases, including HMDB and METLIN. PCA and OPLS-DA, performed using the ropls package in R, examined sample groupings and potential biomarkers. VIP scores ranked discriminatory features, with scores >1 deemed significant. MetaboAnalyst (version 5.0) conducted MSEA and pathway analysis to explore the biological implications of identified metabolites. The

Student's t-test (two-tailed) assessed statistical significance for normally distributed data or the Mann-Whitney U test for non-normally distributed data, applying an FDR correction for multiple testing. Fold change (FC) analysis compared metabolite relative abundance between groups. Metabolites with P-values < 0.05, FC > 1, and VIP > 1 were considered significantly altered. The ComplexHeatmap package in R generated hierarchical clustering analysis (HCA) and heatmaps, visualizing significantly altered metabolite patterns. Pearson's or Spearman's correlation coefficients determined correlations between metabolite levels and other variables based on data distribution. A p-value < 0.05 indicated statistical significance.

## Reporting summary

Further information on research design is available in the Nature Portfolio Reporting Summary linked to this article.

## Data availability

The publicly available National Inpatient Sample (NIS) database (https://hcup-us.ahrq.gov/db/nation/nis/nisdbdocumentation.jsp), Lipid Maps database (https://lipidmaps.org/), HMDB database (https://hmdb.ca/) and KEGG compound database (https://www.genome.jp/kegg/compound/) were used in this study. The data supporting the findings of this study are available within the article and its Supplementary Information files and source data file. The dataset containing raw intensities of metabolites is available through Figshare under [https://doi.org/10.6084/m9.figshare.25062287.v1]. Source data are provided with this paper.

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

## Acknowledgements

We thank Mr. Bo Cai, Ms. Xi Wu and their team from Boston Scientific Corporation for technique support in IVUS image analysis. This study was supported by National Natural Science Foundation of China (82070858) to Z.L; Youth Scientific Research and Innovation Team Program of Shaanxi Province (2022-SLRH-LJ-014) to Z.L; QinChuangYuan TCM Innovation and Research Project (2022-QCYZH-016) to Z.L; SPPH Scientific Research Supporting Project (2021BJ-02) to Z.L; SPPH Scientific Research Supporting Project (2021LJ-03) to J.W.; SPPH Scientific Research Supporting Project (2021BJ-06) to S.P.; SPPH Scientific Research Supporting Project (2023JY-13, 2022YJY-39) to W.X.; Basic Scientific Research Operating Expenses of Xi'an Jiaotong University (xzy012022131) to W.X.; China Postdoctoral Science Foundation (2022M722576) to W.X.; Shaanxi Provincial Postdoctoral Foundation first-class support (No.127) to W.X.; Xi 'an Innovation Ability Support Plan (23YXYJ0187) to W.X.

## Author contributions

All authors made substantial contributions to the conception or design of the study; the acquisition, analysis, or interpretation of data; or drafting or revising the paper. All authors approved the paper. All authors agree to be personally accountable for individual contributions and to ensure that questions related to the accuracy or integrity of any part of the work are appropriately investigated, resolved, and the resolution documented in the literature. Z.L., S.P., Y.Z. and J.W. designed the study and analyses. X.W., L.Z., J.L., Y.M. and G.G. performed the experiments. C.Q. and L.Z. performed the data analysis. C.L., Y.G. and Y.Y. performed the human cohort study. The paper was prepared by Z.L. and X.W. with input from all authors. All authors have reviewed and agreed upon the content of the manuscript before submission.

## Competing interests

The authors declare no competing interests.
