## [Peer Review File · Nature Communications]

Palmitic Acid in Type 2 Diabetes Mellitus Promotes Atherosclerotic Plaque Vulnerability via Macrophage Dll4 SignalingREVIEWER COMMENTS

Reviewer #1 (Remarks to the Author):

The paper is very interesting and fairly novel. The Authors attempt to tackle a challenging yet worthy problem in understanding the complex interplay between metabolic disorders in T2D and atherosclerotic plaque vulnerability. The Authors examine the functional role of palmitic acid (PA) accumulation on plaque vulnerability, the relationship between elevated PA levels and plaque instability (with a focus on the role of Dll4 during M1 macrophage polarization), and examine how Dll4-expressing macrophages can induce VSMC senescence via the Notch pathway.

However, whilst the data appear to be of a generally good standard there is a need for more focus of the paper and the employment of further techniques to address some of the questions raised and fully elucidate the mechanisms of action.

-Immunoblots must be cropped in a way that retains information about antigen size and antibody specificity. The cropped images must retain sufficient area around the band(s) of interest, including the positions of at least one actual molecular weight marker above and one below the band(s). Please prepare each run including at least two lanes for each experimental group. Please avoid overexposed bands (especially in the loading controls). Moreover, to ensure reproducibility by other laboratories, the manufacturer name and catalog number(s) of all the antibodies used should be clearly specified in the material and method section. Please provide uncropped gels as supplementary material showing visible MW markers.

-The effects on VSMC phenotypic switch should be tested.

Patients:

-How was the smoking status defined? The authors should distinguish between current and former smokers.

-What is tobacco abuse?

Mouse model:

-Please show the specificity and efficiency of the KO.

-Figure 3C: please provide at least 2 representative pics per group.

In vitro assays:

-Senescence should be confirmed using the "CellEvent" Senescence Green Reagent.

-Data on cell pH should be provided.

-Methods:

The "contact and non-contact co-culturing models of macrophages and VSMCs" is not well described and no reference is provided (most likely to protect the double-blind model of the review process): please include more details on these protocols in the methods in order to allow the reproducibility of these assays by other labs.

The discussion in its present form somehow fails to interpret the data in the context of what is known in the field: it sounds somehow redundant, as it largely summarizes again data already presented in the Results without placing them in the proper scientific context. Moreover, clinical implications should be better outlined.

Overall, the paper is mainly descriptive and focused on its conclusion, not adequately acknowledging the limitations of the study. The strengths and limitations of the study should be deeply addressed, taking into account sources of potential bias or imprecision: Discuss both direction and magnitude of any potential bias.

Minor concerns:

- The color palette is not red-green color blind friendly. Please redo the figures without red and green on the same figure.
- The font size used in some figures is too small.
- Why is the text in the supplementary file formatted as centered?

Reviewer #2 (Remarks to the Author):

Patients with Type 2 Diabetes Mellitus (T2DM) are more susceptible to atherosclerotic cardiovascular diseases. The detailed mechanisms remain unclear. In this study, the authors identified a significant link that elevated serum levels of palmitic acid regulate the TLR4/ERK/FOXC2 pathway, leading to Dll4 and Notch signaling activation. Future studies targeting the modulation of Dll4 or the interruption of the TLR4/ERK/FOXC2 pathway could pave the way for novel intervention methods. Some major concerns are listed below:

1. In the first cohort study, the authors included 2318638 T2DM and 5357761 Non-T2DM

cases to evaluate the difference in cardiovascular events between the two groups. Results showed that hyperlipidemia and hypertension contribute the most to the occurrence of acute coronary syndrome and acute cerebral infarction, more significantly than T2DM. The authors should carefully explain the findings and link to the second Prospective Cohort, mainly Fatty Acids and Conjugates play may play critical roles in hypertension regardless of T2DM. "T2DM increased the risk of acute cerebral-cardiovascular events" is already known and can not be a remarkable finding.

2. In the second cohort study of "Chronic Stable Coronary Atherosclerotic Disease Prospective Cohort," it will be beneficial if the authors provide the baseline and endpoint metabolomic data. Besides, 424, 616, and 1037 metabolites within the ESI-, ESI+, 542, and COMB were identified as significant factors. The authors need to describe the study design of human metabolomic profiling, notably when to collect samples (indicated in Figure S1) and how they define significant markers using Progenesis QI software (signal intensity, fold change, and p-value). Notably, present the signal intensity, fold change, and p-value of Fatty Acids and Conjugates. The number of Annotated metabolites is less important. The authors should narrow down and only present the most significant 10-30 changed metabolites instead of all so that readers can see the trends of each metabolite (Figure 2C). The comparison of changed metabolites was between Non-T2DM and T2DM but not with/without cardiovascular disease; how can the authors define Fatty Acids and Conjugates as the most significant contributors to atherosclerotic cardiovascular diseases?

3. In the second cohort study, did the authors analyze the difference in sphingolipids between groups? Sphingolipids play important roles in immune cell maturation, activation and determine cell fates. Besides, lysophosphatidylcholine is one of the most abundant serum lipid species that trigger inflammatory signaling. Why the authors only focused on fatty acids but not other lipid contributors? Again, the number of metabolites is not the most important; the abundance/concentration and foldchange are critical. Did the authors isolate the immune cells from plasma or macrophages from tissue and check the expression levels of Dll4?

4. The apoE knockout mouse was not an ideal animal model for this study (APOE KO + Mice received palmitic acid treatment were fed with an HFD containing 5% palmitic acid 16 (customized by 221 Tropic Animal Feed High-Tech Co.) for 20 weeks"). The study design was different from the pathological conditions of humans (T2DM to cardiovascular diseases).

Besides, recent reports showed that at eight weeks of age, the plasma lipids already show significantly different from C57BL/6. Sphingolipids and lysophosphatidylcholines are highly elevated. A high-fat diet is another contributor to cardiovascular diseases. *Int. J. Mol. Sci.* 2023, 24(8), 6956. There are too many contributing factors; the authors should have proper controls; otherwise can not conclude "Palmitic Acid Promotes Atherosclerotic Plaque Vulnerability in Type 2 Diabetes Mellitus." This study lacks controls.

Minor concerns:

1. Figure 3A: HFD (high-fat diet) instead of FHD?
2. Page 1, line 15: metabolomics instead of metabonomics?

Reviewer #3 (Remarks to the Author):

Overall, this paper presents a comprehensive and well-executed study that provides valuable insights into the pathophysiology of atherosclerosis in patients with Type 2 Diabetes Mellitus (T2DM). The authors have successfully identified a strong association between elevated levels of palmitic acid and the vulnerability of atherosclerotic plaques in T2DM patients. The study also uncovers the role of Dll4 in macrophages and the TLR4/ERK/FOXC2 pathway in mediating these effects, providing a novel understanding of the molecular mechanisms involved.

The use of both human subjects and murine models strengthens the validity of the findings, and the integration of metabolomic profiles from both human and murine samples is a commendable approach. The authors have also done an excellent job in presenting their data, with clear figures and thorough statistical analysis.

1. What are the noteworthy results?

The study presents several noteworthy results:

Firstly, The researchers found that serum palmitic acid concentrations were significantly higher in T2DM subjects compared to non-T2DM individuals. This increase was associated with a decrease in the thickness of the fibrous cap of atherosclerotic plaques, indicating a higher vulnerability to plaque rupture. This finding is significant as it suggests a potential biomarker for assessing cardiovascular risk in T2DM patients. Secondly, The study

discovered that the expression of Dll4 in macrophages was upregulated in response to palmitic acid administration. This upregulation was associated with the activation of the Notch pathway in vascular smooth muscle cells (VSMCs), leading to their senescence and contributing to plaque vulnerability. This result is noteworthy as it uncovers a novel mechanism of plaque vulnerability in T2DM. Thirdly, the researchers demonstrated that the knockout of Dll4 in macrophages led to a decrease in plaque burden and an increase in collagen content, indicating improved plaque stability. This result is significant as it suggests a potential therapeutic target for reducing cardiovascular risk in T2DM patients. Fourthly, The study found that palmitic acid upregulated the expression of Dll4 in macrophages via the activation of the TLR4/ERK/FOXC2 pathway. This result is important as it provides insights into the molecular mechanisms underlying the effects of palmitic acid on plaque vulnerability.

2. Will the work be of significance to the field and related fields? How does it compare to the established literature?

The findings of this study are of significant relevance to the field of cardiovascular disease research, particularly in the context of Type 2 Diabetes Mellitus (T2DM). The study provides a novel understanding of the molecular mechanisms underlying the increased vulnerability of atherosclerotic plaques in T2DM patients, which is a critical aspect of the disease that contributes to the high cardiovascular risk associated with T2DM.

The identification of elevated palmitic acid levels as a key factor associated with plaque vulnerability in T2DM patients is a noteworthy contribution to the existing body of literature. This finding aligns with previous studies that have highlighted the role of dysregulated lipid metabolism in T2DM and its impact on cardiovascular health. However, this study goes a step further by elucidating the role of Dll4 in macrophages and the TLR4/ERK/FOXC2 pathway in mediating the effects of palmitic acid, thereby providing a more detailed understanding of the pathophysiological processes involved.

The study also stands out in its use of both human subjects and murine models, and the integration of metabolomic profiles from both these sources. This approach not only strengthens the validity of the findings but also enhances their translational potential.

In comparison to the established literature, this study provides a more comprehensive and detailed understanding of the mechanisms underlying plaque vulnerability in T2DM. The

findings have important implications for the development of new therapeutic strategies aimed at reducing cardiovascular risk in T2DM patients, and as such, are likely to stimulate further research in this area.

3. Does the work support the conclusions and claims, or is additional evidence needed?

The work presented in this study appears to support the conclusions and claims made by the authors. The study is well-designed and includes both human and murine models, which strengthens the validity of the findings. The authors have used a range of experimental techniques, including metabolomic profiling, cell co-culture models, and genetic manipulation (DII4 knockout), to investigate the role of palmitic acid and DII4 in plaque vulnerability in T2DM. The results from these experiments provide strong evidence for the conclusions drawn by the authors. I think it would be beneficial to show β -SA-Gal stain of aortic root of animals to support the change of AS phenotype better.

Overall, while the evidence presented in this study is robust and supports the authors' conclusions, additional studies in larger human cohorts and different animal models would further strengthen the claims made in this study. Authors could claim it in Limitation Section.

4. Are there any flaws in the data analysis, interpretation and conclusions? Do these prohibit publication or require revision?

The authors of this study have conducted a comprehensive and rigorous analysis of their data, employing a variety of techniques including metabolomic profiling, animal model studies, and cell culture experiments. They have also used robust statistical methods to validate their findings. However, there are a few areas where the study could potentially be improved. It's unclear whether the authors controlled for potential confounding factors such as diet, physical activity, and medication use, which could influence both T2DM status and atherosclerotic plaque vulnerability. While the authors have done a commendable job of correlating their findings in murine models with human data, it would be beneficial to further discuss the limitations of these models and how they might impact the interpretation of the results.

5. Is the methodology sound? Does the work meet the expected standards in your field?

The methodology employed in this study appears to be sound and meets the expected standards in the field of metabolic research and atherosclerosis. The authors have used a combination of human cohort studies, animal models, and in vitro experiments to investigate the role of palmitic acid and Dll4 in atherosclerotic plaque vulnerability in T2DM. This multi-pronged approach is commendable as it allows for the validation of findings across different experimental settings, thereby enhancing the robustness of the conclusions drawn. The authors could provide more details on the selection criteria for the human subjects included in the study, as well as any steps taken to control for potential confounding factors. In their in vitro experiments, the authors could provide more details on the cell culture conditions and the methods used to induce M1 polarization in macrophages. A demonstration of induction of M1 polarization would be better to clarify this point.

6. Is there enough detail provided in the methods for the work to be reproduced?

The authors have provided a reasonable amount of detail in their methods section, which forms the basis for the reproducibility of their work. The use of established databases for metabolomic analysis, the description of the generation of macrophage-lineage conditional Dll4 knock-out atherosclerotic mice, and the explanation of the in vitro experiments all contribute to the reproducibility of the study.

Responses to the reviewers:

Reviewer #1

The paper is very interesting and fairly novel. The Authors attempt to tackle a challenging yet worthy problem in understanding the complex interplay between metabolic disorders in T2D and atherosclerotic plaque vulnerability. The Authors examine the functional role of palmitic acid (PA) accumulation on plaque vulnerability, the relationship between elevated PA levels and plaque instability (with a focus on the role of Dll4 during M1 macrophage polarization), and examine how Dll4-expressing macrophages can induce VSMC senescence via the Notch pathway.

However, whilst the data appear to be of a generally good standard there is a need for more focus of the paper and the employment of further techniques to address some of the questions raised and fully elucidate the mechanisms of action.

1) Immunoblots must be cropped in a way that retains information about antigen size and antibody specificity. The cropped images must retain sufficient area around the band(s) of interest, including the positions of at least one actual molecular weight marker above and one below the band(s). Please prepare each run including at least two lanes for each experimental group. Please avoid overexposed bands (especially in the loading controls). Moreover, to ensure reproducibility by other laboratories, the manufacturer name and catalog number(s) of all the antibodies used should be clearly specified in the material and method section. Please provide uncropped gels as supplementary material showing visible MW markers.

Response: We appreciate your feedback on the presentation of our immunoblots. In response, we have reprocessed our immunoblot images to retain essential antigen size and antibody specificity information, ensuring the inclusion of molecular weight markers on the left side. Each run now contains two lanes of targeted proteins for each experimental group. Additionally, we updated the contents of “immunofluorescent stains” and “Western Blots” in Materials and Methods section to specify the manufacturer name and catalog number for all antibodies used. Uncropped gels with

visible molecular weight markers have been added as supplementary material (Figure S19 and Figure S20) to enhance transparency and reproducibility.

2) The effects on VSMC phenotypic switch should be tested.

Response: Thank you for your valuable suggestion. We assessed the phenotypic switch of VSMCs by examining osteopontin (OPN) expression, a marker for synthetic VSMCs, using immunofluorescent staining. Our data revealed elevated OPN expression in VSMCs co-cultured with palmitic acid-treated macrophages. However, this increase was attenuated in VSMCs co-cultured with Dll4-silenced macrophages (Figure S22). These findings have been integrated into the Materials and Methods as well as Results sections.

3) How was the smoking status defined? The authors should distinguish between current and former smokers.

Response: Smoking status is typically defined by categorizing individuals into different groups based on their tobacco or smoking habits. It is commonly assessed in healthcare settings and research studies to understand an individual's relationship with smoking, as it has significant implications for health outcomes and treatment planning. The most common categories used to define smoking status include: 1) Current Smoker: This category includes individuals who are actively smoking tobacco products at the time of assessment or data collection. 2) Former Smoker (Ex-Smoker): Former smokers are individuals who used to smoke in the past but have quit smoking at the time of assessment. This category may also include individuals who use alternative tobacco products, such as electronic cigarettes or smokeless tobacco, even if they have quit traditional cigarette smoking. 3) Never Smoker: Never smokers are individuals who have never used tobacco products in their lifetime. They have never initiated smoking, and their lungs and health are typically considered to be in a healthier state compared to current or former smokers. Thanks so much for your kind reminder, we have added the number of former smokers and never smokers in the Table 2.

4) What is tobacco abuse?

Response: In the International Classification of Diseases, 10th Revision (ICD-10), tobacco abuse is classified under the code F17. This code specifically refers to "Mental and behavioral disorders due to use of tobacco," and it includes conditions related to the harmful use of tobacco products. In this study of NIS cohort, we used the ICD 10 Code to define the tobacco abuse. ICD-10 provides a range of codes within this category to specify different manifestations and complications of tobacco use, such as nicotine dependence (F17.2), tobacco use disorder (F17.2), withdrawal state (F17.2), and other related conditions. The specific code used depends on the diagnosis and the severity of the tobacco-related issue. It's important to note that ICD-10 codes are primarily used for medical and billing purposes and can help healthcare providers and organizations document and track tobacco-related disorders for clinical and statistical purposes.

5) Please show the specificity and efficiency of the KO.

Response: In light of your feedback, we conducted additional experiments to demonstrate the specificity and efficiency of the Dll4 knockout (KO) in our model. We isolated the three primary cell types from the aorta: macrophages, vascular smooth muscle cells (VSMCs), and endothelial cells. Subsequent Western blot analysis was performed to assess Dll4 expression in these isolated cells. Our results unequivocally showed that Dll4 expression was effectively silenced in the isolated macrophages. In contrast, aortic VSMCs and endothelial cells retained their Dll4 expression, underscoring the specificity of our macrophage-targeted KO approach (Figure S15). These findings have been integrated into the Materials and Methods and Results sections to provide a comprehensive understanding of the KO's specificity and efficiency.

6) Figure 3C: please provide at least 2 representative pics per group.

Response: Thank you for your feedback on Figure 3C. We acknowledge the importance of providing a comprehensive representation for clarity and validation. We added additional representative figures in Figure S7.

7) Senescence should be confirmed using the "CellEvent" Senescence Green Reagent.

Response: Thank you for the recommendation to utilize the "CellEvent" Senescence Green Reagent for a more definitive confirmation of senescence. We acknowledge the importance of employing multiple methods to validate cellular senescence. In response to this comment, we conducted additional experiments using the CellEvent™ Senescence Green Detection Kit (ThermoFisher) to further validate the senescence status of our cells. The results from this assay were consistent with our previous findings of SA- β -gal stains, further strengthening our conclusions regarding cellular senescence. We have incorporated these new data into the revised manuscript and provided the corresponding images in Figure 6F and Figure S17.

8) Data on cell pH should be provided.

Response: β -galactosidase (β -gal) is a lysosomal enzyme that is expressed in all cells and that has an optimum pH of 4.0. During senescence, lysosomes increase in size and, consequently, senescent cells accumulate β -gal. The SA- β -gal could be distinguished from the acidic (lysosomal) β -galactosidase activity by using a citric acid/sodium phosphate buffer at pH 6.0. Our protocols of SA- β -gal staining strictly followed previous reports [1]. The CellEvent Senescence Green probe stain was used to confirm the senescence detected by SA- β -gal stain. We re-wrote the content of senescence detection entitled "Assessment of VSMCs senescence" in Materials and Methods section, with a special focus on the importance of pH in senescence detection with SA- β -gal. Though pH is crucial for SA- β -gal activity, the overall intracellular pH is not typically used as a direct marker of cellular senescence. We still implemented additional experiments detecting intracellular pH using fluorescent staining with a cytosolic pH indicator pHrodo Red to detect intracellular pH (pHi) in VSMCs. The results which were demonstrated in Figure S23 would be benefit in ensuring any observed changes in SA- β -gal activity are due to senescence and not due to dramatic alterations in intracellular pH that could affect the enzyme's activity.

Reference

[1] Protocols to detect senescence-associated beta-galactosidase (SA-beta-gal) activity, a biomarker of senescent cells in culture and in vivo [J]. Nat Protoc, 2009, 4(12): 1798-1806

9) The "contact and non-contact co-culturing models of macrophages and VSMCs" is not well described and no reference is provided (most likely to protect the double-blind model of the review process): please include more details on these protocols in the methods in order to allow the reproducibility of these assays by other labs.

Response: Thank you for pointing out the need for a more detailed description of our co-culturing models. We appreciate the emphasis on reproducibility. We have refined and expanded the methodology section pertaining to the co-culturing systems. The updated details can be found under the "co-culturing" subsection in the Materials and Methods section. Additionally, for further clarity and context, we have cited our previous work as a reference. A minor modification on Figure 6C was also made.

10) The discussion in its present form somehow fails to interpret the data in the context of what is known in the field: it sounds somehow redundant, as it largely summarizes again data already presented in the Results without placing them in the proper scientific context. Moreover, clinical implications should be better outlined.

Response: Thank you for your constructive feedback regarding the Discussion section of our manuscript. We acknowledge the importance of contextualizing our findings within the broader scientific landscape and appreciate the emphasis on drawing clearer clinical implications. We re-wrote the whole Discussion section to better integrate our findings with existing knowledge in the field. We made efforts to minimize redundancy and avoid reiterating data already presented in the Results section. Instead, we have focused on providing a more comprehensive interpretation of our results in the context of current scientific understanding. Additionally, a Limitations and Perspectives section was added to the end of the manuscript. Potential clinical implications were stated in this section.

11) Overall, the paper is mainly descriptive and focused on its conclusion, not adequately acknowledging the limitations of the study. The strengths and limitations of the study should be deeply addressed, taking into account sources of potential bias or imprecision: Discuss both direction and magnitude of any potential bias.

Response: Thank you for your constructive feedback regarding the acknowledgment of the study's limitations. We concur that a comprehensive understanding of the study's strengths and limitations is crucial for interpreting the results in the broader context of the field. In response to your comment, we have added a "Limitations and Perspectives" section to the manuscript. We appreciate your guidance in enhancing the depth and clarity of our manuscript.

12) The color palette is not red-green color blind friendly. Please redo the figures without red and green on the same figure.

Response: Thank you for highlighting the importance of accessibility in our figures, especially for those with red-green color blindness. We have avoided using red and green in the same figure and have opted for contrasting colors that are distinguishable by individuals with color vision deficiencies.

13) The font size used in some figures is too small.

Response: Thank you for pointing out the font size issue in some of our figures. We understand the importance of ensuring that all elements within our figures are clearly readable. We have now adjusted the font size in the relevant figures to enhance readability.

14) Why is the text in the supplementary file formatted as centered?

Response: We apologize for the oversight in the formatting of the supplementary file. The text was inadvertently centered during the document preparation process. We have now corrected this and reformatted the supplementary file with the standard left-aligned text.

Reviewer #2

1) In the first cohort study, the authors included 2318638 T2DM and 5357761 Non-T2DM cases to evaluate the difference in cardiovascular events between the two groups. Results showed that hyperlipidemia and hypertension contribute the most to the occurrence of acute coronary syndrome and acute cerebral infarction, more significantly than T2DM. The authors should carefully explain the findings and link to the second Prospective Cohort, mainly Fatty Acids and Conjugates play may play critical roles in hypertension regardless of T2DM. "T2DM increased the risk of acute cerebral-cardiovascular events" is already known and can not be a remarkable finding.

Response: Prior studies have indicated that hypertension and hyperlipidemia were risk factors for acute coronary syndrome and cerebral infarction, and in our first cohort, we once again validated these findings. Unstable plaques represent a primary cause of acute coronary syndrome and cerebral infarction. Therefore, in our second cohort, we included hypertension, low-density lipoprotein (which are risk factors for the acute coronary and cerebral infarction), PA (which showed significant differences between diabetic and non-diabetic groups) and other factors in the multi-regression analysis. The results demonstrated that hypertension and low-density lipoprotein were not associated with plaque instability, while PA was correlated with plaque instability. This suggests that even after accounting for the traditional influences of hypertension and low-density lipoprotein on plaques, PA continues to play a significant role in plaque instability.

In order to confirm whether the Fatty Acids and Conjugates play an important role in the hypertension, we compared the PA in the hypertension group and non-hypertension group, and the multi-logistic was used to determine the relationship between the PA and the hypertension. The results demonstrated that there was no difference of the serum concentration of palmitic acid in hypertension subjects compared to non-hypertension individuals [(99.01±13.48) vs. (96.23±12.89) µg/L, P=0.309]. Then we included the age, gender, BMI, PA, LDL, smoking and T2DM in the multi-logistic analysis, the result demonstrated that PA had no relationship with hypertension [OR 0.978 (0.940-1.016); P=0.255].

2) In the second cohort study of "Chronic Stable Coronary Atherosclerotic Disease Prospective Cohort," it will be beneficial if the authors provide the baseline and endpoint metabolomic data. Besides, 424, 616, and 1037 metabolites within the ESI-, ESI+, 542, and COMB were identified as significant factors. The authors need to describe the study design of human metabolomic profiling, notably when to collect samples (indicated in Figure S1) and how they define significant markers using Progenesis QI software (signal intensity, fold change, and p-value). Notably, present the signal intensity, fold change, and p-value of Fatty Acids and Conjugates. The number of Annotated metabolites is less important. The authors should narrow down and only present the most significant 10-30 changed metabolites instead of all so that readers can see the trends of each metabolite (Figure 2C). The comparison of changed metabolites was between Non-T2DM and T2DM but not with/without cardiovascular disease; how can the authors define Fatty Acids and Conjugates as the most significant contributors to atherosclerotic cardiovascular diseases?

Response:

Thank you for your valuable feedback and for highlighting specific areas that required further clarification in our manuscript. We appreciate your suggestion regarding the provision of both baseline and endpoint metabolomic data. For the "Chronic Stable Coronary Atherosclerotic Disease Prospective Cohort," the metabolomic data we provided pertains to the baseline. The primary objective of our follow-up for this cohort was to document the incidence rate of MACE events, mainly conducted through interviews or phone follow-ups. Regrettably, we were unable to collect blood specimens during the follow-up, which means we couldn't provide endpoint metabolomic data for this cohort. We have taken your feedback into account and added the specific timing for blood sample collection in Figure S1. To clarify, blood samples were gathered within a 12-hour window following invasive coronary evaluation. We define significant metabolites primarily based on their fold change, VIP and p-value. For subjects with biological replicates, the method of combining the fold change, P-value and VIP value

was used to screen the differential metabolites. The screening criteria were $FC > 1$, P value < 0.05 , and $VIP > 1$. We revised related descriptions in the Statistic section. For a comprehensive view of all metabolites within "Fatty Acids and Conjugates", we added data concerning signal intensity, \log_2 fold change, and p -value in Table S4.

We concur with your perspective on the presentation of metabolites. To provide a clearer and more focused view, we have removed the heatmaps from Figure2, Figure3, and Figure4 that displayed all differential metabolites. Instead, we have now presented the top 30 significantly altered metabolites in Figure2E, Figure3I, and Figure4E. In our second human cohort, all enrolled patients were diagnosed with stable angina pectoris through coronary angiography, indicating that all participants had cardiovascular diseases. Our approach was as follows: 1) Firstly, we categorized these patients into non-T2DM and T2DM groups. 2) Secondly, we compared the differential metabolites between the non-T2DM and T2DM groups. 3) Thirdly, within the significantly altered metabolites, we sought those associated with plaque instability features, both radiologically (IVUS acoustic characteristics) and clinically (follow-up MACE events). Through HMDB and KEGG clustering analysis, fatty acid metabolism disruption emerged prominently. Further analysis using Lipid Maps revealed that among the significantly altered metabolites, the most frequent changes were observed in "Fatty Acids and Conjugates". In our Human-mouse convergence model, we correlated all selected candidate metabolites (including Fatty Acyls and other lipids) with plaque instability features (intravascular echographic and clinical characteristics). After reviewing existing literature on the association between these metabolites and cardiovascular diseases, we identified palmitic acid as a key metabolite.

We would like to clarify our approach in this study. As a high-throughput research method, metabolomics allows us to screen a vast array of significantly altered metabolites. Our intention was to harness this capability to identify as many differential metabolites as possible and subsequently correlate them with clinical characteristics. While we acknowledge the importance of absolute change of intensity, we believe that

the subsequent correlation analysis could play a pivotal role in identifying the targeted metabolite.

We appreciate your feedback on the phrasing concerning "Fatty Acids and Conjugates as the most significant contributors to atherosclerotic cardiovascular diseases". We recognize the need for precision in our statements and have revised it to: "The significant number of altered metabolites within 'Fatty Acids and Conjugates' drew our attention, prompting a deeper investigation into their relationship with atherosclerotic cardiovascular diseases associated with plaque instability." Your feedback has been valuable in refining our manuscript, and we are grateful for the opportunity to clarify these points.

3) In the second cohort study, did the authors analyze the difference in sphingolipids between groups? Sphingolipids play important roles in immune cell maturation, activation and determine cell fates. Besides, lysophosphatidylcholine is one of the most abundant serum lipid species that trigger inflammatory signaling. Why the authors only focused on fatty acids but not other lipid contributors? Again, the number of metabolites is not the most important; the abundance/concentration and foldchange are critical. Did the authors isolate the immune cells from plasma or macrophages from tissue and check the expression levels of Dll4?

Response:

Thank you for your valuable suggestions. In our second cohort study, we did perform a comprehensive lipidomic analysis, which included sphingolipids. However, for the sake of clarity and focus in the manuscript, we primarily highlighted the findings related to fatty acids, given their pronounced alterations and potential implications in our study context. We acknowledge the significant role of sphingolipids in immune cell functions and their potential impact on atherosclerosis. Based on your suggestion, we have now included a description (in Limitation and Perspective section) detailing the differences in sphingolipids between the groups (TableS4). However, we found that the change of lysophosphatidylcholine was not significant. Moreover, the linear correlation analysis

did not identify any significant correlations between dramatically altered sphingolipids (namely galabiosylceramide, C-6 NBD-dihydro-Ceramide, Dimethylsphingosine and C16 Sphingosine) with characteristics of atherosclerotic plaque vulnerability.

Thank you for raising the question regarding our focus on fatty acids over other lipid contributors. In our human-mouse convergence model, we analyzed all significantly changed metabolites. Through a linear correlation analysis between these metabolites and the atherosclerotic plaque vulnerability indicator (plaque fibrous cap thickness), we identified only eight metabolites (1-pentadecanoyl-glycero-3-phosphate, 11-Deoxycortisol, cholyasparagine, palmitic acid, icosanoic acid, isolinderenolide, Ornithylamphotericin methyl ester, and tilmicosin) that were significantly correlated with the thickness of the plaque fibrous cap (Table S5). Subsequently, a 180-day follow-up study was conducted to determine the association of these metabolites with major adverse cardiovascular events (MACE). Among these, only palmitic acid was found to be associated with MACE, while the other seven metabolites showed no significant correlation (Figure S12). Moreover, out of these eight metabolites, four (1-pentadecanoyl-glycero-3-phosphate, 11-Deoxycortisol, Icosanoic acid, and Tilmicosin) were annotated by the Lipid Maps database, with two of them belonging to the "fatty acid" category. Given these findings, our study honed in on fatty acids, particularly emphasizing palmitic acid, a saturated fatty acid, due to its significant association with both plaque fibrous cap thickness and MACE. This detailed analysis and rationale have been incorporated into the Results section under the subsection "Association of Elevated Palmitic Acid Levels in T2DM with Atherosclerotic Plaque Vulnerability."

We appreciate the reviewer's emphasis on the significance of metabolite abundance/concentration and fold change. In our study, we meticulously analyzed the metabolites that exhibited significant alterations across the human cohort, mice, and the human-mouse convergence model. Based on the magnitude of the log₂ fold change, we highlighted the top 30 most significantly altered metabolites, as presented in Figure2E, Figure3I, and Figure4E. Leveraging the HMDB and KEGG databases, we directed our

attention specifically towards lipid metabolism disturbances and further refined our analysis using the Lipid Maps database. This analysis revealed that the most pronounced changes among the significant metabolites belonged to the "Fatty acyls" category, specifically "fatty acids and conjugates" (as depicted in Figure2G-H, Figure3L-M, Figure4I-J). Consequently, we detailed the significantly changed metabolites within "fatty acids and conjugates" based on their log₂ fold change (Figure2I, Figure3N, Figure4K). Furthermore, we conducted a comprehensive correlation analysis on all significantly altered metabolites in the human-mouse convergence model, irrespective of whether they were related to lipid metabolism. This approach led us to identify palmitic acid as a metabolite of particular interest. Our selection methodology was twofold: firstly, identifying all metabolites with significant changes, and secondly, correlating these significantly altered metabolites with clinical indicators to pinpoint our target metabolites. In this study, our primary focus was to identify metabolites that exhibited significant changes and further explore the metabolic pathways in which these metabolites predominantly participate. This approach combining our correlation analyses with clinical indicators, aided us in further selecting and pinpointing a key metabolite with specific clinical significance. While we concur that metabolite abundance is paramount, it is essential to interpret it in conjunction with clinical implications. In subsequent research, we plan to delve deeper into the metabolites with the highest abundance to investigate their association with specific clinical scenarios.

We would like to clarify that in our study, we did not obtain atherosclerotic tissue samples from human subjects, nor did we isolate immune cells from human blood samples. Consequently, we were unable to assess the expression levels of Dll4 in human atherosclerotic plaque macrophages or blood immune cells. However, we did isolate macrophages from the atherosclerotic plaques of animal models, specifically from AS and AS+DM animals. After cell identification, we analyzed the Dll4 expression levels within these macrophages. As depicted in Figure S24, the expression levels of Dll4 in macrophages from AS+DM plaques were significantly elevated compared to those

from AS plaques.

4) The apoE knockout mouse was not an ideal animal model for this study (APOE KO + Mice received palmitic acid treatment were fed with an HFD containing 5% palmitic acid 16 (customized by 221 Tropic Animal Feed High-Tech Co.) for 20 weeks"). The study design was different from the pathological conditions of humans (T2DM to cardiovascular diseases). Besides, recent reports showed that at eight weeks of age, the plasma lipids already show significantly different from C57BL/6. Sphingolipids and lysophosphatidylcholines are highly elevated. A high-fat diet is another contributor to cardiovascular diseases. *Int. J. Mol. Sci.* 2023, 24(8), 6956. There are too many contributing factors; the authors should have proper controls; otherwise can not conclude "Palmitic Acid Promotes Atherosclerotic Plaque Vulnerability in Type 2 Diabetes Mellitus." This study lacks controls.

Response:

We appreciate the reviewer's concerns regarding the use of the apoE knockout mouse model combined with a high-fat diet to simulate atherosclerosis. Our primary intention behind this approach was to investigate the specific impact of palmitic acid on atherosclerotic plaque vulnerability. The combination of apoE knockout mice with high-fat diet is an ideal model for generating pronounced atherosclerotic plaques. In such a model, the effects of palmitic acid on plaque instability become more discernible. This approach was taken to ensure that we could effectively observe the impact of palmitic acid on significant plaque vulnerabilities. Given the multifaceted metabolic alterations associated with diabetes, this model allowed us to focus solely on the effects of palmitic acid, eliminating other potential confounding factors inherent to a diabetic state.

The reviewer rightly pointed out the significant role of a high-fat diet in atherosclerosis. To address this, we introduced additional control groups: apoE knockout mice and Dll4 conditional knockout mice without the high-fat diet but stimulated solely with palmitic acid. As depicted in Figure 5, while the high-fat diet significantly increased plaque

burden, it did not markedly influence collagen deposition within the plaque, Dll4 expression in plaque macrophages, or NICD1 expression levels in plaque vascular smooth muscle cells. Interestingly, even in the absence of a high-fat diet, palmitic acid stimulation in apoE knockout mice exhibited similar effects, including increased plaque burden, reduced collagen content, elevated Dll4 expression in macrophages, and increased NICD1 expression in vascular smooth muscle cells.

Based on our findings from both human and animal studies, we believe it is reasonable to conclude that "Palmitic acid is elevated in T2DM, which promotes atherosclerotic plaque vulnerability." However, we acknowledge that our study design might not capture the full complexity of the interplay between palmitic acid, atherosclerosis, and T2DM. A more fitting model for our study would ideally involve animals that exhibit both diabetes and elevated endogenous levels of palmitic acid. However, the development and validation of such a model require further exploration, especially concerning the sources of endogenous palmitic acid. For instance, the dysregulation of gut microbiota in diabetic conditions could be a significant contributor to increased endogenous palmitic acid levels. Delving deeper into these mechanisms will be instrumental in establishing a more appropriate animal model for our research objectives. We added the statement in the Limitation and Perspectives section.

Minor concerns:

1) Figure 3A: HFD (high-fat diet) instead of FHD?

Response: We are very sorry for the typo. We revised it in Figure3A.

2) Page 1, line 15: metabolomics instead of metabonomics?

Response: We are very sorry for the typo. We revised it.

Reviewer #3

Overall, this paper presents a comprehensive and well-executed study that provides valuable insights into the pathophysiology of atherosclerosis in patients with Type 2

Diabetes Mellitus (T2DM). The authors have successfully identified a strong association between elevated levels of palmitic acid and the vulnerability of atherosclerotic plaques in T2DM patients. The study also uncovers the role of Dll4 in macrophages and the TLR4/ERK/FOXC2 pathway in mediating these effects, providing a novel understanding of the molecular mechanisms involved.

The use of both human subjects and murine models strengthens the validity of the findings, and the integration of metabolomic profiles from both human and murine samples is a commendable approach. The authors have also done an excellent job in presenting their data, with clear figures and thorough statistical analysis.

1) What are the noteworthy results?

The study presents several noteworthy results:

Firstly, The researchers found that serum palmitic acid concentrations were significantly higher in T2DM subjects compared to non-T2DM individuals. This increase was associated with a decrease in the thickness of the fibrous cap of atherosclerotic plaques, indicating a higher vulnerability to plaque rupture. This finding is significant as it suggests a potential biomarker for assessing cardiovascular risk in T2DM patients. Secondly, The study discovered that the expression of Dll4 in macrophages was upregulated in response to palmitic acid administration. This upregulation was associated with the activation of the Notch pathway in vascular smooth muscle cells (VSMCs), leading to their senescence and contributing to plaque vulnerability. This result is noteworthy as it uncovers a novel mechanism of plaque vulnerability in T2DM. Thirdly, the researchers demonstrated that the knockout of Dll4 in macrophages led to a decrease in plaque burden and an increase in collagen content, indicating improved plaque stability. This result is significant as it suggests a potential therapeutic target for reducing cardiovascular risk in T2DM patients. Fourthly, The study found that palmitic acid upregulated the expression of Dll4 in macrophages via the activation of the TLR4/ERK/FOXC2 pathway. This result is important as it provides insights into the molecular mechanisms underlying the effects of palmitic acid on plaque vulnerability.

Response: Thank you for the comprehensive summary and recognition of the pivotal findings from our research. We appreciate your insights and believe that these findings significantly advance the current understanding of cardiovascular complications associated with T2DM.

2) Will the work be of significance to the field and related fields? How does it compare to the established literature?

The findings of this study are of significant relevance to the field of cardiovascular disease research, particularly in the context of Type 2 Diabetes Mellitus (T2DM). The study provides a novel understanding of the molecular mechanisms underlying the increased vulnerability of atherosclerotic plaques in T2DM patients, which is a critical aspect of the disease that contributes to the high cardiovascular risk associated with T2DM.

The identification of elevated palmitic acid levels as a key factor associated with plaque vulnerability in T2DM patients is a noteworthy contribution to the existing body of literature. This finding aligns with previous studies that have highlighted the role of dysregulated lipid metabolism in T2DM and its impact on cardiovascular health. However, this study goes a step further by elucidating the role of Dll4 in macrophages and the TLR4/ERK/FOXO2 pathway in mediating the effects of palmitic acid, thereby providing a more detailed understanding of the pathophysiological processes involved. The study also stands out in its use of both human subjects and murine models, and the integration of metabolomic profiles from both these sources. This approach not only strengthens the validity of the findings but also enhances their translational potential. In comparison to the established literature, this study provides a more comprehensive and detailed understanding of the mechanisms underlying plaque vulnerability in T2DM. The findings have important implications for the development of new therapeutic strategies aimed at reducing cardiovascular risk in T2DM patients, and as such, are likely to stimulate further research in this area.

Response: Thank you for your insightful feedback. Your appreciation of our dual approach, utilizing both human subjects and murine models, is heartening. We believe

that this integrative methodology not only bolsters the robustness of our findings but also paves the way for their potential clinical application.

3) Does the work support the conclusions and claims, or is additional evidence needed?

The work presented in this study appears to support the conclusions and claims made by the authors. The study is well-designed and includes both human and murine models, which strengthens the validity of the findings. The authors have used a range of experimental techniques, including metabolomic profiling, cell co-culture models, and genetic manipulation (D114 knockout), to investigate the role of palmitic acid and D114 in plaque vulnerability in T2DM. The results from these experiments provide strong evidence for the conclusions drawn by the authors. I think it would be beneficial to show β -SA-Gal stain of aortic root of animals to support the change of AS phenotype better.

Overall, while the evidence presented in this study is robust and supports the authors' conclusions, additional studies in larger human cohorts and different animal models would further strengthen the claims made in this study. Authors could claim it in Limitation Section.

Response: Thank you for your insightful feedback and the recommendation to further validate the changes in the atherosclerosis phenotype. In response to your suggestion, we have conducted supplementary experiments to assess cellular senescence in the aortic root of animals. As depicted in the revised Figure 5F and Figure S17, we employed the "CellEvent" Senescence Green Reagent for staining specimens from different animal groups. Our results indicate that palmitic acid stimulation indeed elevates the fluorescence levels of CellEvent Green, suggesting its role in promoting cellular senescence. While β -SA-Gal staining is frequently utilized in *in vitro* studies, the CellEvent Senescence Green Reagent offers superior performance for *in vivo* studies involving animal tissue specimens. Consequently, we opted for this method to analyze cellular senescence in the aortic root. We concur with your perspective on the potential benefits of additional studies in larger human cohorts and different animal models. We recognize the importance of such studies in further validating and

strengthening our claims. We note this point in the Limitation and Perspectives section of our manuscript.

4) Are there any flaws in the data analysis, interpretation and conclusions? Do these prohibit publication or require revision?

The authors of this study have conducted a comprehensive and rigorous analysis of their data, employing a variety of techniques including metabolomic profiling, animal model studies, and cell culture experiments. They have also used robust statistical methods to validate their findings. However, there are a few areas where the study could potentially be improved. It's unclear whether the authors controlled for potential confounding factors such as diet, physical activity, and medication use, which could influence both T2DM status and atherosclerotic plaque vulnerability. While the authors have done a commendable job of correlating their findings in murine models with human data, it would be beneficial to further discuss the limitations of these models and how they might impact the interpretation of the results.

Response: In our cohort, patients were categorized into non-T2DM and T2DM groups based on their T2DM diagnosis. The baseline data for these groups, including medication use, is detailed in Table 2. An essential point we failed to emphasize initially is that all patients included in our cohort had never been previously diagnosed with coronary heart disease or T2DM. Consequently, none of the patients had been on any glucose-lowering medications prior to our study. Given the aforementioned point that all participants had not been previously diagnosed with coronary heart disease or T2DM, there were no significant differences in diet and physical activity between the two groups. Both groups were naive to any interventions or lifestyle modifications related to these conditions. We added these related descriptions in “Study population” under “Prospective cohort registry study” section. We concur with your observation regarding the limitations of murine models. While murine models offer valuable insights, they inherently possess physiological and metabolic differences from humans. In our "Limitations and Perspectives" section, we have highlighted the challenges associated with the ApoE^{-/-} mouse model and the need for cautious interpretation when

extrapolating results to human conditions. We will further emphasize the potential discrepancies between murine and human pathophysiology and how they might influence the interpretation of our findings.

5) Is the methodology sound? Does the work meet the expected standards in your field? The methodology employed in this study appears to be sound and meets the expected standards in the field of metabolic research and atherosclerosis. The authors have used a combination of human cohort studies, animal models, and in vitro experiments to investigate the role of palmitic acid and Dll4 in atherosclerotic plaque vulnerability in T2DM. This multi-pronged approach is commendable as it allows for the validation of findings across different experimental settings, thereby enhancing the robustness of the conclusions drawn. The authors could provide more details on the selection criteria for the human subjects included in the study, as well as any steps taken to control for potential confounding factors. In their in vitro experiments, the authors could provide more details on the cell culture conditions and the methods used to induce M1 polarization in macrophages. A demonstration of induction of M1 polarization would be better to clarify this point.

Response: Thank you for your constructive feedback regarding our methodology, particularly concerning the cell culture conditions and the induction of M1 polarization in macrophages. In response to your comment, we have provided a more detailed description of our cell culture conditions and the methods employed for inducing M1 polarization in the "cell treatments and apoptosis assay" section. Specifically, primary macrophages were treated with varying concentrations of palmitic acid, namely 0, 0.1, 0.2, and 0.4 mmol/L, to observe its effect on M1 macrophage polarization. The results of these treatments, which demonstrate the influence of palmitic acid on M1 macrophage polarization, are presented in Figure S18. We have made revisions to the "Dll4 Silencing Attenuates Notch Signaling-Mediated VSMC Senescence by Suppressing Palmitic Acid-Induced Macrophage TLR4 Pathway Activation" in Results section.

6) Is there enough detail provided in the methods for the work to be reproduced?

The authors have provided a reasonable amount of detail in their methods section, which forms the basis for the reproducibility of their work. The use of established databases for metabolomic analysis, the description of the generation of macrophage-lineage conditional Dll4 knock-out atherosclerotic mice, and the explanation of the in vitro experiments all contribute to the reproducibility of the study.

Response: Thank you for acknowledging the level of detail provided in our methods section. We appreciate your positive feedback and will continue to prioritize transparency and detail in our future work.

REVIEWER COMMENTS

Reviewer #1 (Remarks to the Author):

The Authors addressed all the Reviewers' concerns in a satisfactorily manner.

Reviewer #2 (Remarks to the Author):

This manuscript still has numerous errors and can NOT be published in its present form.

Major concerns are listed below.

1. Please ensure that palmitic acid presents a "negative correlation" or "positive correlation" with the vulnerability of plaques. The first few words of the title are "Palmitic Acid Promotes Atherosclerotic Plaque Vulnerability". It indicates that a higher concentration of Palmitic Acid (PA) could damage the vascular system. However, on lines 14 to 16 of page 1, the authors emphasized that "Key findings a strong negative correlation between serum palmitic acid identified by metabolomics and the vulnerability of plaques". The grammar is wrong, and the meaning is contrary to the title of this manuscript. The authors must check the terms and make them consistent in the text content. Also, check Figures 1M and 1O and the figure legends.

2. In the title words "Palmitic Acid Promotes Atherosclerotic Plaque Vulnerability" PA seems to be the most crucial atherogenic factor, according to the authors; however, the interpretation of metabolomic data could be different. In other words, why is the PA not present in Figure 2E? That data means PA is not one of the most significantly different metabolites between groups. In Figures 2G, 2H, and 2I, the authors point out that fatty acyls play essential roles in disease. Unfortunately, in 2E and 2I, some other factors are more significant than PA. The authors are not able to neglect those factors presented in the figures. Can the authors fix the missing links here?

3. In Figures 2I, 3N, and 4K, the fold changes of PA between groups are not much; how do the authors decide the concentration for the animal study in Figure 5? How do the authors explain the PA concentration in normal physical conditions and in vascular diseases? Is the

concentration for the animal study similar to the disease group of humans?

4. Mostly, the authors have responded to the reviewers' comments properly. However, regarding the signal intensity issue, the authors should be conscientious and think about the implications. For omics data, we always need to set a cut-off value far above the background noise. We should ignore those contents with extremely low signal intensity, i.e., <3000 or <1000. That is because of the limitation of the mass spectrometry Xevo G2-XS QTOF. For example, the highest factor in Table S4 is 3-carboxy-4-methyl-5-pentyl-2-furanpropanoic acid in the control group, which is 21542.62. On the contrary, if the authors still want to analyze (S)-2-Aceto-2-hydroxybutanoate, 3,5-hexadienoic acid, in which the intensities are less than 10, what is the reason? I believe the signal intensity of the most abundant lipid could be greater than 1,000,000. The dominant metabolites are more significant. Besides, with the low abundance metabolites such as (S)-2-Aceto-2-hydroxybutanoate, the compound identification scores in the Progenesis Q1 software should be low as well, no matter whether they use the METLIN database or a self-constructed library (Biomark). The above-mentioned has become the major concern of this study. For example, the compound C-6 NBD-dihydro- Ceramide in Table S6 is an artificial derivative and does NOT exist in animals.

Other minor comments are:

1. Suppl file lines 24-36 and lines 39-43 on pages 1-2, information is repeated.
2. Figures S5, S8-11..etc. All figures need to check the font size and resolution.
3. ApoE knockout: apoE should be *Italic*. All genes should be *Italic*. Check all the text content and Figures. For example, check Figures 2 and S13.
4. Check grammar and spelling in all the text content and Figures. For example, on line 949, The data supporting the findings of this study are available within the article and "ites" Supplementary Information files and "Figshare" with the identifier... The authors have to revise the content carefully.

Reviewer #3 (Remarks to the Author):

Regarding my comments, I have thoroughly reviewed the revisions made to the manuscript.

I am pleased to note that the authors have addressed all the concerns raised comprehensively and have made the necessary modifications. Upon reviewing the revised manuscript, I was particularly impressed with the newly added research findings. The CellEvent senescence staining in the aortic root provides significant value to the study, effectively reflecting the in vivo phenotypic changes and offering deeper insights. Additionally, I commend the authors for introducing the Limitation section. This section clearly delineates the existing limitations of the current study and potential directions for future research. These modifications have rendered the manuscript more rigorous and insightful. I am now in favor of its publication.

Responses to the reviewers:

Reviewer #1

The Authors addressed all the Reviewers' concerns in a satisfactorily manner.

Response: We sincerely appreciate your positive feedback and are gratified to learn that our revisions have satisfactorily addressed the concerns raised. Your constructive comments have been invaluable in enhancing the quality and clarity of our manuscript. We are committed to maintaining the high standards of research and presentation, and your acknowledgment of our efforts is greatly encouraging.

Reviewer #2

1) Please ensure that palmitic acid presents a "negative correlation" or "positive correlation" with the vulnerability of plaques. The first few words of the title are "Palmitic Acid Promotes Atherosclerotic Plaque Vulnerability". It indicates that a higher concentration of Palmitic Acid (PA) could damage the vascular system. However, on lines 14 to 16 of page 1, the authors emphasized that "Key findings a strong negative correlation between serum palmitic acid identified by metabolomics and the vulnerability of plaques". The grammar is wrong, and the meaning is contrary to the title of this manuscript. The authors must check the terms and make them consistent in the text content. Also, check Figures 1M and 1O and the figure legends.

Response: Thank you for your insightful observations and for highlighting the inconsistencies in our manuscript regarding the correlation between palmitic acid and plaque vulnerability. We appreciate your attention to detail, which has helped us improve the clarity and accuracy of our work. We have revised the sentence on lines 14-16 to accurately reflect the findings of our study. The revised sentence now reads: "Key findings indicate a strong positive correlation between serum palmitic acid identified by metabolomics and the vulnerability of plaques." We have also reviewed and ascertained the legends for Figure 4O and Figure 4M to ensure they accurately

represent our findings. Figure 4O, the Kaplan-Meier survival curve demonstrates that subjects in the higher palmitic acid group exhibited a greater incidence of major adverse cardiovascular events (MACE) during the 180-day follow-up, compared to the lower palmitic acid group. This observation supports the conclusion that higher serum palmitic acid levels are associated with increased plaque vulnerability. Similarly, in Figure 4M, the Spearman linear regression analysis shows a significant negative correlation between serum palmitic acid concentration and coronary plaque fibrous cap thickness which is indicative of increased plaque vulnerability.

2) In the title words "Palmitic Acid Promotes Atherosclerotic Plaque Vulnerability" PA seems to be the most crucial atherogenic factor, according to the authors; however, the interpretation of metabolomic data could be different. In other words, why is the PA not present in Figure 2E? That data means PA is not one of the most significantly different metabolites between groups. In Figures 2G, 2H, and 2I, the authors point out that fatty acyls play essential roles in disease. Unfortunately, in 2E and 2I, some other factors are more significant than PA. The authors are not able to neglect those factors presented in the figures. Can the authors fix the missing links here?

Response: Thank you for your constructive feedback, particularly for emphasizing the importance of a comprehensive interpretation of our metabolomic data. We appreciate the opportunity to clarify why palmitic acid (PA) was chosen as a focal point of our study, especially in comparison to other metabolites that may exhibit more significant changes. While PA was not among the top 30 significantly altered metabolites in Figure 2E, we prioritized it in our research due to its specific relevance to the vulnerability of atherosclerotic plaques, particularly in the context of Type 2 Diabetes Mellitus (T2DM). Our emphasis on PA stems from its strong correlation with plaque instability, a connection that was substantiated through our comprehensive analysis. As demonstrated in the schematic figure (**Figure R1**) below, our approach to identifying metabolites related to the instability of atherosclerotic plaques involved several steps: 1) Conducting metabolomic studies in populations with the disease phenotype (stable coronary artery disease, with or without diabetes); 2) Establishing similar disease

phenotype animal models (atherosclerotic animals, with or without diabetes) and conducting metabolomic studies; 3) Overlapping the metabolomic data from human cohorts and animal models to build a convergent model, thereby narrowing down the range of candidate metabolites; 4) Utilizing databases like HMDB, KEGG, and Lipidmaps to identify the most significantly altered metabolite types; 5) Most crucially, investigating the relationship between these candidate metabolites and the instability of atherosclerotic plaques. We conducted extensive analyses (correlating metabolites with both intravascular echographic and clinical features of vulnerable plaques), ultimately identifying PA as our target metabolite. We acknowledge that there are many metabolites with changes more significant than PA (as shown in Figures 2E, 2G, 2H, and 2I). However, the relevance of these metabolites to our target disease phenotype (atherosclerotic plaque instability) is of paramount importance. We hope this response would address your concerns and clarify our rationale for selecting and focusing on PA in our study. We are grateful for your insights, which have undoubtedly strengthened the manuscript.

Figure R1: Methodological flowchart for target metabolites screening

3) In Figures 2I, 3N, and 4K, the fold changes of PA between groups are not much; how do the authors decide the concentration for the animal study in Figure 5? How do the authors explain the PA concentration in normal physical conditions and in vascular diseases? Is the concentration for the animal study similar to the disease group of humans?

Response: We are grateful for the opportunity to clarify these aspects of our study. While the fold change of PA may not appear notably high, it is statistically significant. We conducted a comprehensive correlation analysis of all differentially expressed metabolites identified in the human-mouse convergence model with our research focus (plaque instability), considering both intravascular echographic and clinical features. PA emerged as the only metabolite showing significant correlation with both, thus demonstrating the strongest biological relevance rather than other candidate

metabolites identified by the human-mouse convergence model (TableS5 and FigureS12). The choice of PA concentration in our animal studies was informed by previous research (reference #16), which indicated that a High-Fat Diet (HFD) containing 5% PA effectively induces an unstable atherosclerotic plaque phenotype in animals. Actually, we have also conducted a pilot experiment to validate this concentration. As illustrated **Figure R2** below, a HFD containing 5% PA significantly induced plaque instability, whereas a 7% concentration, although effective on inducing plaque instability, led to severe liver dysfunction, as indicated by markedly elevated transaminase levels. Therefore, we opted for the 5% concentration. In our study, we did not measure PA levels in the serum of healthy individuals. The mechanisms underlying elevation of PA levels is complex and may be associated with multiple factors related to the pathophysiology of Type 2 Diabetes Mellitus, such as lipid metabolism disorders and gut microbiota dysbiosis. We hypothesize that vascular disease might be a pathological outcome of abnormal PA accumulation in the body, rather than a direct cause of elevated PA levels. This hypothesis warrants further investigation and evidence, which could be an intriguing direction for future research. Our use of a PA-enriched HFD in animal models was intended to observe the impact of PA on atherosclerotic plaque instability. This model was not designed to mimic vascular disease or diabetes states per se. Moreover, while the trend of metabolite level changes in disease states should be consistent between humans and rodent models, it is unlikely that the concentration levels of specific metabolites will be entirely exact same across human and rodent physiology.

Figure R2: Optimization of Palmitic Acid Concentration for Inducing Plaque Instability in Animal Models

A: Sirius Red staining of the aortic root in animals fed with a high-fat diet (HFD) and diets supplemented with varying concentrations of palmitic acid (PA). B: Bar graph representing the relative positive collagen area in aortic root plaques stained with Sirius Red at different PA concentrations. C: Bar graph showing the levels of the liver enzyme alanine aminotransferase (ALT) in animals subjected to different concentrations of PA. The scale bar indicates a length of 400 μ m (* P<0.05; *** P<0.001, n=3)

4) Mostly, the authors have responded to the reviewers' comments properly. However, regarding the signal intensity issue, the authors should be conscientious and think about the implications. For omics data, we always need to set a cut-off value far above the background noise. We should ignore those contents with extremely low signal intensity, i.e., <3000 or <1000. That is because of the limitation of the mass spectrometry Xevo G2-XS QTOF. For example, the highest factor in Table S4 is 3-carboxy-4-methyl-5-pentyl-2-furanpropanoic acid in the control group, which is 21542.62. On the contrary, if the authors still want to analyze (S)-2-Aceto-2-hydroxybutanoate, 3,5-hexadienoic acid, in which the intensities are less than 10, what is the reason? I believe the signal

intensity of the most abundant lipid could be greater than 1,000,000. The dominant metabolites are more significant. Besides, with the low abundance metabolites such as (S)-2-Aceto-2-hydroxybutanoate, the compound identification scores in the Progenesis QI software should be low as well, no matter whether they use the METLIN database or a self-constructed library (Biomark). The above-mentioned has become the major concern of this study. For example, the compound C-6 NBD-dihydro- Ceramide in Table S6 is an artificial derivative and does NOT exist in animals.

Response: We are grateful for the reviewer's constructive feedback regarding the signal intensity issue in our omics data. We acknowledge the importance of setting a cut-off value to reduce the potential influence of noise in our analyses. In line with your valuable recommendation, we have established a cut-off value to exclude noise interference. Based on guidance from Waters Corporation (defining 'signal' as the peak height of the chromatographic peak of interest, and 'noise' as the average intensity of a continuous section of the mass chromatogram over a 0.3-minute window at the start of evaluation), we revisited our human and animal serum metabolomics data. As demonstrated in **Figure R3**, We calculated the average noise intensity for the internal standard (2-Aminochlorobenzyl acid) during the first 0-0.3 minutes of evaluation. The average noise intensities for the human and animal metabolomics were 13.97 and 17.69, respectively. Accordingly, we set the detection cut-off value based on a signal-to-noise ratio (S/N) of 3, as recommended in previous investigations (References [1, 2]), resulting in cut-off values of 42 for human and 53 for animal metabolomics analysis. Utilizing these cut-off values, we reanalyzed the metabolomics data for humans, animals, and the human-mouse convergence model. This reevaluation necessitated modifications in the Results section (specifically in Figure 2A, 2B, 2C, 2D, 2F, 2G, 2H, Figure 3F, 3G, 3H, 3J, 3K, 3L, 3M, Figure 4A, 4B, 4C, 4D, 4F, 4H, 4I, 4J, and Figures S3, S4, S5, S8, S9, S10, S11). We believe that the application of these cut-off values significantly bolsters the credibility of our study. We have now included a description of the cut-off value setting in the Methods section. Further, upon re-examining Table S4 with these cut-off values, we deleted 11 metabolites that fell below the threshold. In Table S6, we identified two sphingolipids, C-6 NBD-dihydro-Ceramide and

Dimethylsphingosine, with signal intensities of 15.13 and 18.05, respectively, which were below our cut-off value and have thus been removed. We appreciate the opportunity to refine our approach and enhance the scientific rigor of our study. This revision not only improves the quality of our current research but also provides invaluable learning for our future endeavors.

Figure R3: Noise Analysis in Metabolomic Studies Using Internal Standard

A: Mass spectrometry chromatogram of the internal standard, 2-Aminochlorobenzyl acid, during the human metabolomic analysis. B: Enlarged view of the noise profile from 0 to 0.3 minutes in the human metabolomic analysis. C: Mass spectrometry chromatogram of the internal standard, 2-Aminochlorobenzyl acid, during the animal metabolomic analysis. D: Enlarged view of the noise profile from 0 to 0.3 minutes in the animal metabolomic analysis.

References

- [1] Kao CY, Kuo PY, Liao HW. Untargeted Microbial Exometabolomics and Metabolomics Analysis of *Helicobacter pylori* J99 and jhp0106 Mutant [J]. *Metabolites*, 2021, 11(12): 808
- [2] Hao L, Wang J, Page D, et al. Comparative Evaluation of MS-based Metabolomics Software and Its Application to Preclinical Alzheimer's Disease [J]. *Sci Rep*, 2018, 8(1): 9291

Other minor comments are:

- 1) Suppl file lines 24-36 and lines 39-43 on pages 1-2, information is repeated.

Response: Thank you for pointing out the mistakes in the supplementary file. We have

revised these sections to eliminate the mistakes and ensure clarity and conciseness in our presentation.

2) Figures S5, S8-11..etc. All figures need to check the font size and resolution.

Response: We are grateful for your observation regarding the font size and resolution in Figures S5, S8-11, and others in the supplementary material. We have thoroughly reviewed these figures and made necessary adjustments to the font size and resolution.

3) ApoE knockout: apoE should be Italic. All genes should be Italic. Check all the text content and Figures. For example, check Figures 2 and S13.

Response: We appreciate your meticulous attention to the formatting of gene names in our manuscript and figures. In accordance with your suggestion, we have thoroughly reviewed the entire text and all figures, ensuring that all gene names.

4) Check grammar and spelling in all the text content and Figures. For example, on line 949, The data supporting the findings of this study are available within the article and "ites" Supplementary Information files and "Figshare" with the identifier... The authors have to revise the content carefully.

Response: Thank you for pointing out the need for a thorough grammar and spelling check throughout our manuscript and figures. We have conducted a comprehensive review of the entire text, including all figures, to identify and correct any grammatical errors and spelling mistakes.

Reviewer #3

Regarding my comments, I have thoroughly reviewed the revisions made to the manuscript. I am pleased to note that the authors have addressed all the concerns raised comprehensively and have made the necessary modifications. Upon reviewing the revised manuscript, I was particularly impressed with the newly added research findings. The CellEvent senescence staining in the aortic root provides significant value

to the study, effectively reflecting the in vivo phenotypic changes and offering deeper insights. Additionally, I commend the authors for introducing the Limitation section. This section clearly delineates the existing limitations of the current study and potential directions for future research. These modifications have rendered the manuscript more rigorous and insightful. I am now in favor of its publication.

Response: Thank you for your thoughtful and detailed review of our revised manuscript. We are deeply gratified to hear that the modifications and additions we made, particularly the inclusion of the CellEvent senescence staining data and the comprehensive Limitations section, have met with your approval and enhanced the overall value of our study.

REVIEWERS' COMMENTS

Reviewer #2 (Remarks to the Author):

The authors have adequately addressed some of the comments. However, the responses to reviewer comments #2 and #3 are unacceptable.

1. The HFD with 5%PA is not equal to elevated PA metabolite in the blood of diabetes (patients/animals). The mechanisms and the implications of blood PA are different from feeding.

Author's reply: "In our study, we did not measure PA levels in the serum of healthy individuals. The mechanisms underlying elevation of PA levels is complex and may be associated with multiple factors related to the pathophysiology of Type 2 Diabetes Mellitus, such as lipid metabolism disorders and gut microbiota dysbiosis. We hypothesize that vascular disease might be a pathological outcome of abnormal PA accumulation in the body, rather than a direct cause of elevated PA levels. This hypothesis warrants further investigation and evidence, which could be an intriguing direction for future research."

2. The authors still cannot logically explain why PA stands out from the thirty most significant metabolites. If that idea (PA vs. plaque instability) is from a previous study, then using an omic survey is unnecessary. Moving all the omic data to the supplementary section is an option.

Author's reply: "While PA was not among the top 30 significantly altered metabolites in Figure 2E, we prioritized it in our research due to its specific relevance to the vulnerability of atherosclerotic plaques, particularly in the context of Type 2 Diabetes Mellitus (T2DM)."

3. The authors failed to demonstrate that the correlation of the thirty factors with plaque instability is less significant than that of PA. Please refer to 2E, 2I, and Table S5. The metabolites were not matched; they need to be clarified. For example, where is the data in Table S5, ethylacrylcarnitine, prostaglandin F2a, N-palmitoyl GABA, 2-amino-8-exo-9,10-epoxy-decanoic acid? Also, if Table S5 is more important, why is it supplementary?

Author's reply: "Our emphasis on PA stems from its strong correlation with plaque instability, a connection that was substantiated through our comprehensive analysis."

REVIEWERS' COMMENTS

Reviewer #2 (Remarks to the Author):

The authors have adequately addressed some of the comments. However, the responses to reviewer comments #2 and #3 are unacceptable.

1. The HFD with 5%PA is not equal to elevated PA metabolite in the blood of diabetes (patients/animals). The mechanisms and the implications of blood PA are different from feeding.

Response: Thank you for your insightful comment regarding the use of a high-fat diet (HFD) with 5% palmitic acid (PA) in our study. We appreciate the opportunity to clarify the purpose and rationale behind our experimental design. We would like to emphasize that the primary objective of employing this animal model was to confirm the impact of PA on plaque instability, as suggested by our preliminary metabolomic analysis and clinical data mining. The model was not intended to mimic the exact pathological environment of diabetes but rather to validate the specific role of PA in plaque vulnerability. Therefore, the concentration of exogenously administered PA does not need to be equivalent to the PA concentration in diabetic patients or animals. The optimum concentration of PA in animal models only needs to meet the requirement of demonstrating its impact of PA on plaque instability. Using a PA-enriched diet to increase circulating PA levels in mice was a strategic approach to specifically study this aspect. Previous investigation utilized similar approach to elevate serum PA [1], which demonstrates that a diet rich in PA can elevate circulating PA levels in mice. This precedent supports the validity of our model as a means to study the effects of elevated PA levels on atherosclerotic plaque vulnerability. We acknowledge that our model does not fully represent the complex metabolic alterations seen in diabetes. However, it serves as a focused approach to isolate and study the effects of PA, which is a critical aspect of our research question.

Reference:

[1] Rui Zhao, Li Cao, Wenjun Gu, et al. Gestational palmitic acid suppresses embryonic GATA-binding protein 4 signaling and causes congenital heart disease [J]. *Cell Rep Med*, 2023, 4(3): 100953

2. The authors still cannot logically explain why PA stands out from the thirty most significant metabolites. If that idea (PA vs. plaque instability) is from a previous study, then using an omic survey is unnecessary. Moving all the omic data to the supplementary section is an option.

Response: We appreciate the opportunity to further clarify why palmitic acid (PA) was selected for focus in our study, despite it not being among the top 30 differential metabolites initially identified.

Firstly, although PA was not among the top 30 differential metabolites, it emerged as a significant molecule through our human-mouse convergence model screening process. This approach was a critical step in identifying metabolites most relevant to our study's

aims.

Secondly, the selection of PA was further reinforced by its clinical relevance, a vital aspect of our research. Notably, PA demonstrated a significant correlation with both the imaging characteristics of plaque instability (as shown in Figure 4M and Figure 4N) and the clinical outcomes associated with unstable plaques, such as increased Major Adverse Cardiac Events (MACE), however, other molecules in 30 differential metabolites were not associated with imaging characteristics of plaque instability or clinical outcomes. Therefore, the dual relevance of PA in terms of imaging features and clinical outcomes significantly underscores its importance in our study. The unique characteristics of PA, specifically its association with both the morphological aspects of plaque instability and the consequent clinical manifestations, aligns closely with our research focus on diabetes-related atherosclerotic plaque instability. This alignment makes PA not just a relevant, but a critical metabolite for our investigation.

Thirdly, the omics data, in conjunction with our clinical findings, provided a comprehensive picture, leading us to PA. And, so far, there have been no relevant reports on the association between PA and plaque instability in human cohort. We believe this integrative approach underscores the rigor of our methodology and the relevance of our findings to the field of diabetes and atherosclerosis. We emphasize that the omics data form the foundation of our metabolite selection process, thereby playing a crucial role in the narrative of our research.

3. The authors failed to demonstrate that the correlation of the thirty factors with plaque instability is less significant than that of PA. Please refer to 2E, 2I, and Table S5. The metabolites were not matched; they need to be clarified. For example, where is the data in Table S5, ethylacrylcarnitine, prostaglandin F2a, N-palmitoyl GABA, 2-amino-8-exo-9,10-epoxy-decanoic acid? Also, if Table S5 is more important, why is it supplementary?

Response: We appreciate the reviewer's attention to detail in our metabolite analysis and the opportunity to clarify our methodology and findings. We would like to clarify that our analysis did not just directly correlate the top 30 human differential metabolites (as shown in Figure 2E) or the human differential metabolites annotated using the Lipidmaps database (as in Figure 2I) with plaque instability. Instead, our approach involved selecting metabolites of interest through a human-mouse convergence model (as detailed in Figure 4). After annotating these selected differential metabolites using the Lipidmaps database (Figure 4G), we conducted a correlation analysis between these annotated metabolites and the imaging characteristics of plaque instability (specifically, fibrous cap thickness). The metabolites that showed a significant correlation with fibrous cap thickness ($P < 0.05$) are listed in Table S5. We then proceeded to analyze the correlation between the metabolites in Table S5 and the clinical features of plaque instability (i.e., the occurrence rate of Major Adverse Cardiac Events, MACE) as shown in Figure S12. This analysis revealed that PA stood out as the only metabolite significantly correlated with both the imaging characteristics and clinical outcomes of plaque instability. It is important to emphasize that Table S5 and Figure 4G correspond with each other, and there is no direct correspondence with Figure 2. Regarding the placement of Table S5 in the supplementary section, given the critical role of this table in our findings, we followed your advice and

integrated it into the main text.